# Genome-scale metabolic modeling reveals SARS-CoV-2-induced metabolic changes and antiviral targets

Kuoyuan Cheng[1,2,†] (iD), Laura Martin-Sancho[3,†] (iD), Lipika R Pal[1] (iD), Yuan Pu[3], Laura Riva[3,‡] (iD), Xin Yin[3,4] (iD), Sanju Sinha[1,2], Nishanth Ulhas Nair[1], Sumit K Chanda[3,*] & Eytan Ruppin[1,5,**] (iD)

## Abstract

Tremendous progress has been made to control the COVID-19 pandemic caused by the SARS-CoV-2 virus. However, effective therapeutic options are still rare. Drug repurposing and combination represent practical strategies to address this urgent unmet medical need. Viruses, including coronaviruses, are known to hijack host metabolism to facilitate viral proliferation, making targeting host metabolism a promising antiviral approach. Here, we describe an integrated analysis of 12 published *in vitro* and human patient gene expression datasets on SARS-CoV-2 infection using genome-scale metabolic modeling (GEM), revealing complicated host metabolism reprogramming during SARS-CoV-2 infection. We next applied the GEM-based metabolic transformation algorithm to predict anti-SARS-CoV-2 targets that counteract the virus-induced metabolic changes. We successfully validated these targets using published drug and genetic screen data and by performing an siRNA assay in Caco-2 cells. Further generating and analyzing RNA-sequencing data of remdesivir-treated Vero E6 cell samples, we predicted metabolic targets acting in combination with remdesivir, an approved anti-SARS-CoV-2 drug. Our study provides clinical data-supported candidate anti-SARS-CoV-2 targets for future evaluation, demonstrating host metabolism targeting as a promising antiviral strategy.

**Keywords** antiviral target; genome-scale metabolic modeling; remdesivir; RNAi screen; SARS-CoV-2
**Subject Categories** Metabolism; Microbiology, Virology & Host Pathogen Interaction; Pharmacology & Drug Discovery
**Mol Syst Biol. (2021) 17: e10260**

## Introduction

The coronavirus disease 2019 (COVID-19), a serious respiratory disease caused by the coronavirus SARS-CoV-2, has evolved into a major pandemic incurring millions of deaths worldwide (all dates as of July 2021; WHO Coronavirus Disease Dashboard, 2021). Despite unprecedented global efforts in response to this serious health threat including abundant studies on the disease biology (e.g., Hoffmann *et al*, 2020, Zhou *et al*, 2020a, reviewed in Tay *et al*, 2020, etc.), preclinical antiviral drug/target screens or predictions (e.g., Daniloski *et al*, 2021, Riva *et al*, 2020, Wei *et al*, 2021, with compiled resources such as Kuleshov *et al*, 2020), and thousands of registered clinical trials on COVID-19 (International Clinical Trials Registry Platform, 2021), therapeutic options remain scarce. Remdesivir, a viral RNA-dependent RNA polymerase inhibitor, represents the only drug approved by the drug regulatory authorities of several countries, including the U.S. Food and Drug Administration (FDA) (Beigel *et al*, 2020), and confers only mild clinical benefits to a subset of COVID-19 patients (WHO Solidarity Trial Consortium *et al*, 2020). 11 different therapies, including the Janus kinase (JAK) inhibitor baricitinib (in combination with remdesivir), and virus-neutralizing antibodies sotrovimab, and casirivimab plus imdevimab, have obtained Emergency Use Authorization (EUA) from the FDA (U.S. Food & Drug Administration, 2021a). Dexamethasone and other corticosteroids have been recommended by the U.S. National Institutes of Health (NIH) for hospitalized patients requiring supplemental oxygen (National Institutes of Health, 2021; RECOVERY Collaborative Group *et al*, 2021). Besides, several SARS-CoV-2 vaccines have been approved or authorized for emergency use in different countries (Dong *et al*, 2020; U.S. Food & Drug Administration, 2021b). Nevertheless, there is still an urgent

---

1   Cancer Data Science Laboratory (CDSL), National Cancer Institute (NCI), National Institutes of Health (NIH), Bethesda, MD, USA
2   Biological Sciences Graduate Program (BISI), University of Maryland, College Park, MD, USA
3   Immunity and Pathogenesis Program, Infectious and Inflammatory Disease Center, Sanford Burnham Prebys Medical Discovery Institute, La Jolla, CA, USA
4   State Key Laboratory of Veterinary Biotechnology, Harbin Veterinary Research Institute, Chinese Academy of Agricultural Sciences, Harbin, China
5   Department of Computer Science, University of Maryland, College Park, MD, USA
    *Corresponding author. Tel: +1 858 795 5241; E-mail: schanda@sbpdiscovery.org
    **Corresponding author. Tel: +1 240 858 3169; E-mail: eytan.ruppin@nih.gov
    †These authors contributed equally to this work
    ‡Present address: Calibr, a Division of The Scripps Research Institute, La Jolla, CA, USA

unmet medical need for the fast identification and development of highly effective anti-COVID-19 therapies.

Viruses are known to "hijack" the host cell metabolism to complete their own intracellular life cycle (Mayer *et al*, 2019), modulating diverse pathways including carbohydrate, lipid, amino acid, and nucleotide metabolism (Sanchez & Lagunoff, 2015; Mayer *et al*, 2019). Coronaviruses including MERS-CoV rearrange cellular lipid profiles upon infection (Yan *et al*, 2019a; Yuan *et al*, 2019b). Recent studies have reported that SARS-CoV-2 also induces changes in numerous metabolic pathways including TCA cycle, oxidative phosphorylation, and lipid metabolism among others in human patient samples (preprint: Ehrlich *et al*, 2020; Gardinassi *et al*, 2020). Notably, counteracting the metabolic demands of viruses including MERS-CoV has been shown to abolish their ability to infect the host cells (Mayer *et al*, 2019; Yuan *et al*, 2019), and the PPARα-agonist fenofibrate can reverse some of the SARS-CoV-2-induced metabolic changes and reduce the viral load (preprint: Ehrlich *et al*, 2020). Therefore, targeting the virus-induced metabolic changes can be a promising novel antiviral strategy (Mayer *et al*, 2019), and can be especially valuable in anti-SARS-CoV-2 drug repurposing to address the current urgent COVID-19 crisis considering that many existing drugs are metabolism-targeting.

Genome-scale metabolic models (GEMs) are *in silico* constraint-based models that comprehensively encompass the cellular network of metabolic reactions, metabolic proteins, and metabolites (Baart & Martens, 2012). GEM analysis has been repeatedly shown to generate accurate predictions and informative hypotheses for metabolism research (Gu *et al*, 2019). Notably, we have previously developed numerous GEM-based algorithms including iMAT (Shlomi *et al*, 2008), which computes genome-wide metabolic fluxes from gene expression profiles, and the metabolic transformation algorithm (MTA; Yizhak *et al*, 2013), which predicts metabolic targets whose inhibition facilitates transformation between specified cellular metabolic states (e.g., from diseased to healthy states). More recently, Valcárcel *et al* (2019) have described a variant of MTA named rMTA with improved performance. Incorporating such high-performance GEM methods in the analysis of data on SARS-CoV-2 infection provides us with a unique opportunity to understand the metabolic demands of SARS-CoV-2 and to systematically predict anti-SARS-CoV-2 targets that counteract the virus-induced metabolic alterations.

Here, we apply GEM algorithms in a comprehensive analysis of 12 published bulk/single-cell RNA-sequencing (RNA-seq/scRNA-seq) and mass spectrometry (MS)-based proteomics datasets on SARS-CoV-2 infection, involving both *in vitro* and human patient samples. We find that metabolic reprogramming represents one of the most consistent molecular changes in SARS-CoV-2 infection besides immune responses, and characterized the complex patterns of metabolic flux alterations. Using rMTA, we predicted anti-SARS-CoV-2 targets that reverse the virus-induced metabolic changes, either as single targets or in combination with remdesivir (the latter using our new RNA-seq data on remdesivir treatment). The predictions are highly enriched for reported anti-SARS-CoV-2 targets identified from various experimental screens, and we further validated a core set of top predicted single targets with an immunofluorescence-based siRNA assay in Caco-2 cells. Our results demonstrate the potential of targeting host metabolism to inhibit viral infection.

# Results

## Integrated analysis of multiple gene expression datasets identifies coherent immune and metabolic changes in SARS-CoV-2 infection

Multiple studies have characterized the gene expression changes during SARS-CoV-2 infection in different *in vitro* and *in vivo* settings. We collected a total of 12 published relevant datasets spanning a wide range of sample types (various cell lines, primary bronchial epithelial cells, nasopharyngeal swab, and bronchoalveolar lavage fluid, i.e., BALF samples from patients) and assay platforms (bulk RNA-seq, scRNA-seq, and MS-based proteomics). These datasets are summarized in Table 1. With each of the datasets, we performed differential expression (DE) analysis comparing the SARS-CoV-2-infected or positive samples to the non-infected control or negative samples (Materials and Methods; Table EV1). For the single-cell datasets, we focused on the airway epithelial cell, which is known as the major virus-infected cell type. Comparing the datasets with a principal component analysis (PCA) plot based on the inverse normal-transformed DE log fold change values (Fig 1A; Materials and Methods) suggests that the cell lines tend to have distinct DE profiles from the patient samples, although different patient datasets exhibit considerable variation depending on sample type and sequencing platform. Such variation is confirmed by comparing the top significant DE genes (FDR < 0.1) from each pair of datasets (Fig 1B; additional robustness analysis in Appendix Fig S1; Materials and Methods). Examining only the top DE genes also appears to mitigate the technical variation across datasets, with reasonable coherence demonstrated by odds ratio median value 1.50 and maximum 5.89 (Fisher's exact test adjusted *P* median 4.56e-6, minimum < 2.22e-16; Fig 1B).

We then performed gene set enrichment analysis (GSEA) (Subramanian *et al*, 2005) on the DE results from each dataset (Table EV2), and further compared the datasets on the pathway level by the significantly enriched pathways (FDR < 0.1; Materials and Methods). Reassuringly, the level of coherence across datasets on the pathway level is even stronger, with a median odds ratio of 4.53 (maximum is infinity followed by 40.73) across pairs of datasets (adjusted *P* median 2.88e-5, minimum < 2.22e-16; Fig 1C). Examining the most consistently enriched pathways across the datasets while giving higher importance to the various *in vivo* patient datasets (Fig 1D; Table EV3; Materials and Methods), we see many up-regulated pathways involved in innate immune response to viral infection, e.g., interferon signaling. Among the pathways involving coherently down-regulated genes upon SARS-CoV-2 infection, we find antigen presentation, as well as numerous pathways spanning many major categories of cellular metabolism, e.g., TCA cycle and the respiratory electron transport, sphingolipid metabolism, glucose metabolism, and *N*-glycan biosynthesis. These may reflect the specific metabolic requirements of SARS-CoV-2 or underlie its pathogenic effects (see Discussion). Visualizing a more complete landscape of metabolic pathway alterations across the datasets reveals further consistent, although weaker, changes (based on GSEA normalized enrichment score, i.e., NES; Fig 1E; Table EV2; Materials and Methods). The major findings above are robust to the DE algorithms used (Appendix Fig S2). These results suggest that besides immune response, metabolic reprogramming represents one of the most

**Table 1.  Summary of the published gene expression datasets on SARS-CoV-2 infection analyzed in this study.**

| Dataset name[a] | Sample type | Sample size[b] | Platform | Reference |
|---|---|---|---|---|
| Vero | Vero E6 cell line | 6 | Bulk RNA-seq | Riva *et al* (2020) |
| NHBE | Primary normal human bronchial epithelial cell | 6 | Bulk RNA-seq | Blanco-Melo *et al* (2020) |
| A549 | A549 human lung adenocarcinoma cell line with exogenous ACE2 expression | 6 | Bulk RNA-seq | Blanco-Melo *et al* (2020) |
| Calu-3 | Calu-3 human lung adenocarcinoma cell line | 6 | Bulk RNA-seq | Blanco-Melo *et al* (2020) |
| 293T | HEK293T human embryonic kidney cell line | 12 | Bulk RNA-seq | Weingarten-Gabby *et al* (2021) |
| Caco-2 | Caco-2 human colorectal adenocarcinoma cell line | 6 | MS-based proteomics | Bojkova *et al* (2020b) |
| Swab.Butler | NP swab samples from human individuals | 580 | Bulk RNA-seq | Butler *et al* (2021) |
| Swab.Lieberman | NP swab samples from human individuals | 484 | Bulk RNA-seq | Lieberman *et al* (2020) |
| BALF | BALF from human individuals | 6 | Bulk RNA-seq | Xiong *et al* (2020b) |
| SC.Liao | BALF from human individuals (epithelial cells were used in analysis) | 13 | scRNA-seq | Liao *et al* (2020) |
| SC.Chua.Basal | NP and bronchial samples from human individuals (basal cells were used in analysis) | 24 | scRNA-seq | Chua *et al* (2020) |
| SC.Chua.Ciliated | NP and bronchial samples from human individuals (ciliated cells were used in analysis) | 24 | scRNA-seq | Chua *et al* (2020) |

BALF, bronchoalveolar lavage fluid; MS, mass spectrometry; NP, nasopharyngeal; RNA-seq, RNA-sequencing; scRNA-seq, single-cell RNA-sequencing.
[a]These are the names used in figure labels throughout the text.
[b]The total number of replicates (virus-infected and control combined) used for analysis in *in vitro* datasets, or the total number of human individuals (patients and controls combined) used for analysis in *in vivo* datasets. In some datasets, only a subset of all the available samples were analyzed.

robust changes induced by SARS-CoV-2 infection across various systems, consistent with the key roles of metabolism in viral infection. We next focused on characterizing the SARS-CoV-2-induced metabolic changes in the infected host cells on the metabolic flux level.

**Genome-scale metabolic modeling (GEM) identifies SARS-CoV-2-induced patterns of metabolic flux changes**

Since gene expression does not necessarily correlate with protein level or enzyme activity and thus may not truthfully reflect metabolic activity (Maier *et al*, 2009), we applied GEM to infer the metabolic fluxes (i.e., rates of all metabolic reactions) across the datasets. Specifically, for each dataset, the iMAT algorithm (Shlomi *et al*, 2008) was applied to the median expression profiles of the control and virus-infected samples to compute the refined metabolic models representative of the two respective groups. Briefly, iMAT uses mixed integer programming to optimally identify high- and low-activity reactions that match the high and low gene expression patterns in a sample-specific manner, thus defining sample-specific model constraints to obtain contextualized models (Shlomi *et al*, 2008). For the base metabolic models, we mainly used the more recent Recon 3D (Brunk *et al*, 2018), but also used Recon 1 (Duarte *et al*, 2007) for increased robustness (Materials and Methods). After obtaining the dataset and sample-specific constrained model with iMAT, the marginal distribution of flux values of each metabolic reaction was obtained by sampling. The flux distributions of the control and infected groups were compared, and reactions with differential fluxes (DF) were identified (Materials and Methods; Table EV4). We again examined the consistency across the datasets,

here on the flux level, by checking the overlap of the top DF reactions between each pair of datasets. Like on the gene expression level, we are assured by the overall reasonable level of coherence of the DF reactions (odds ratio median 2.05, maximum 2.89; adjusted *P* value median 1.45e-11, minimum < 2.22e-16; Fig 2A shows the result for the positive DF reactions, the result is similar for negative DF reactions. We note that the sign of DF represents the direction of flux change with regard to the positive direction of a reaction, which can be reversible, and not the increase or decrease in the absolute flux). Although no reaction shows fully consistent changes across all 12 datasets, we identified a set of most consistently changed reactions across datasets while giving higher importance to the *in vivo* patient datasets (Table EV5A; Materials and Methods), and examined the metabolic pathways they are enriched in with Fisher's exact tests (significant pathways with FDR < 0.1 shown in Fig 2B; Table EV5B). We see that consistent flux changes are found in various noteworthy pathways including metabolite transport (mitochondrial and extracellular), pentose phosphate pathway, hyaluronan metabolism, pyrimidine synthesis, glycine, serine, alanine and threonine metabolism, inositol phosphate metabolism, and fatty acid synthesis, among others. Many of these pathways have been implicated in the infection and life cycle of different viruses including SARS-CoV-2 (Mayer *et al*, 2019; preprint: Bojkova *et al*, 2020a; preprint: Ehrlich *et al*, 2020; Gardinassi *et al*, 2020; Ou *et al*, 2020; Thomas *et al*, 2020; Li *et al*, 2021, see Discussion).

Next, we closely inspect the fluxes within specific pathways by visualizing their alteration patterns overlaid on the metabolic network, for virus-infected vs the control group. For example, the pyrimidine (*de novo*) synthesis pathway mostly contains consistently increased fluxes toward the synthesis of UMP (the precursor

of pyrimidines; Fig 2C), consistent with the nucleic acid synthesis needs of the virus. As examples of pathways with more complex flux change patterns, in the inositol phosphate metabolism pathway, we see increased fluxes converging to phosphatidylinositol 4,5-

bisphosphate (pail45p_hs[c]) and inositol (inost[c]), but decreased fluxes to inositol 1-phosphate (mi1p_DASH_D[c]; Fig 2D); in the fatty acid synthesis pathway, we see that the synthesis and interconversion of different fatty acids show distinct flux changes (Fig 2E).

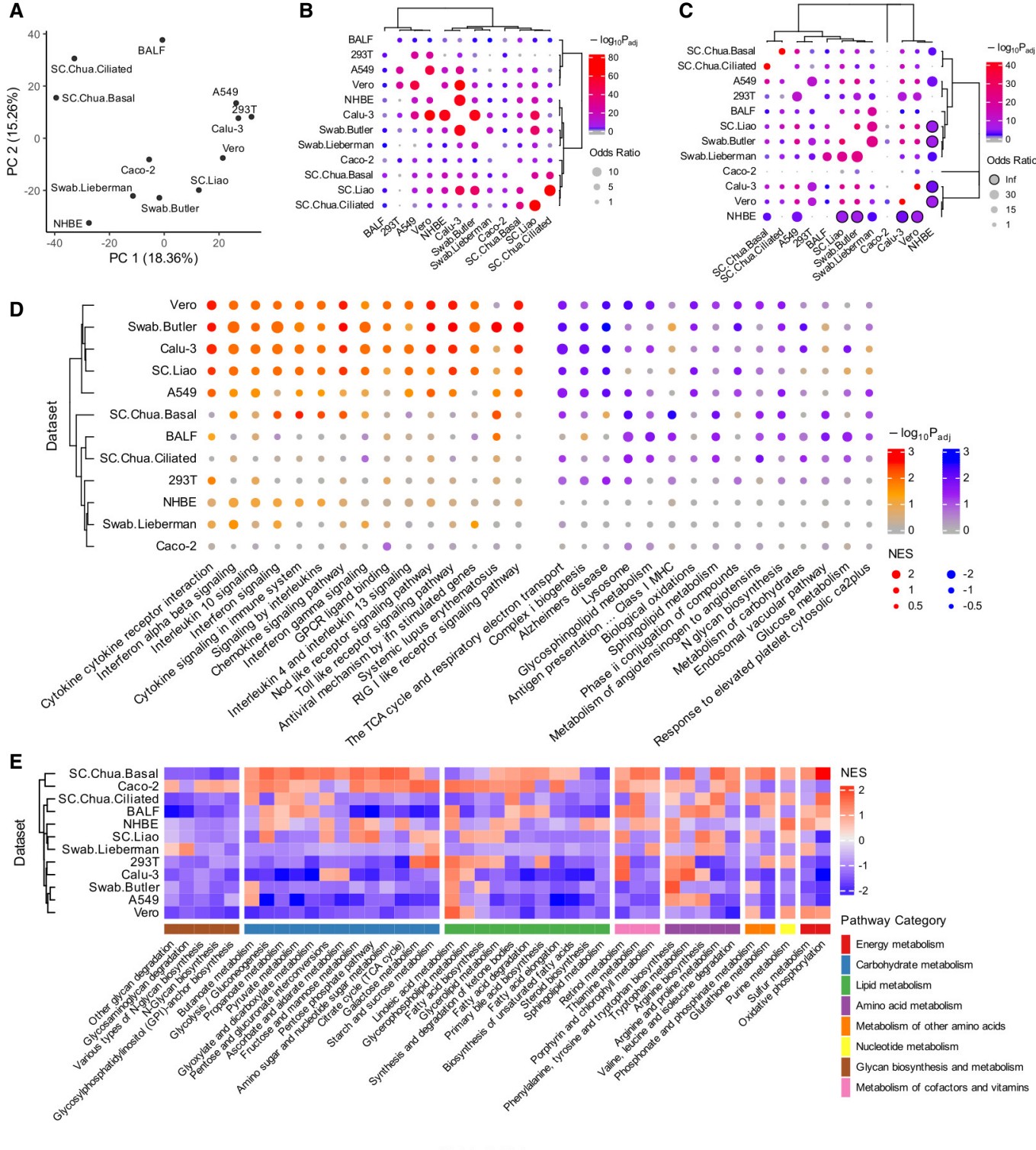

**Figure 1.**

◄

**Figure 1.   Analysis of SARS-CoV-2-induced gene expression changes with 12 published datasets.**

A   PCA plot using the rank-based inverse normal-transformed differential expression (DE) log fold change values (virus-infected compared to control samples) across all the datasets analyzed.

B   Visualization of the overlap of the top significant DE genes (FDR < 0.1) between each pair of datasets analyzed using Fisher's exact tests (Materials and Methods). The dot size corresponds to the effect size of the overlap as measured by odds ratio, and the color corresponds to the negative $\log_{10}$ adjusted one-sided $P$ value (gray means below 0.05).

C   Visualization of the overlap of the top significantly enriched pathways (FDR < 0.1) from the gene set enrichment analysis (GSEA) between each pair of datasets analyzed using Fisher's exact tests (Materials and Methods). The meanings of dot size and color are the same as (B), and dots with black borders correspond to infinity odds ratio.

D   A summary visualization of the GSEA result for the top consistently altered pathways during SARS-CoV-2 infection across the datasets, with more importance given to the various *in vivo* patient datasets (Materials and Methods). The dot color corresponds to the negative $\log_{10}$ adjusted $P$ values from GSEA, with two sets of colors (red-orange and blue-purple) distinguishing up-regulation from down-regulation (positive or negative normalized enrichment scores, i.e., NES); dot size corresponds to the absolute value of NES measuring the strength of enrichment. The left- and right-hand side blocks represent the pathways that tend to be consistently up-regulated and down-regulated in infected vs control samples, respectively; within each block, the pathways are ordered by negative sum of log $P$ values across datasets (i.e., Fisher's method).

E   Heatmap summarizing the landscape of metabolic pathway alterations (based on gene expression) during SARS-CoV-2 across datasets. The heatmap color corresponds to the GSEA NES values (explained above) for KEGG metabolic pathways grouped into major categories. Only the metabolic pathways with FDR < 0.1 enrichment in at least one dataset are included in the heatmap. The dataset labels used in this figure correspond to those given in Table 1.

These highly intricate metabolic programs revealed by the GEM analysis are consistent with many previous reports and possibly reflect the specific metabolic demands of SARS-CoV-2 during its life cycle (see Discussion), which also demonstrates the value of the modeling approach over gene expression-level analyses.

**Prediction of anti-SARS-CoV-2 targets that act via counteracting the virus-induced metabolic changes**

We have demonstrated that SARS-CoV-2 can induce recurrent and complex alterations in host cell metabolism. As was proposed previously, targeting the virus-induced metabolic changes can be an effective antiviral strategy (Mayer *et al*, 2019), which we adopted here to predict anti-SARS-CoV-2 targets. Specifically, we applied the GEM-based rMTA algorithm (Valcárcel *et al*, 2019) to each of our collected datasets to predict metabolic reactions whose knockout (KO) can transform the cellular metabolism from the SARS-CoV-2-infected state to the non-infected normal state, based on both the Recon 3D and Recon 1 models like above for higher robustness (Materials and Methods; Table EV6). Recon 3D results are described below unless otherwise noted. MTA computes a score for each of the metabolic reactions in the cell, and usually, the 10–20% reactions with the highest MTA score contain promising candidate targets (Yizhak *et al*, 2013). We first compared the top 10% MTA-predicted reactions across datasets and found that they have reasonable overlap (odds ratio median 1.83, maximum 6.90, Fisher's exact test adjusted $P$ median 6.95e-14, minimum < 2.2e-16 across all pairs of datasets; Fig 3A). Interestingly, some strong overlaps are seen between certain cell lines and patient datasets, consistent with the recurrent metabolic changes across these datasets as seen above.

To validate these predictions, we collected multiple validation sets of reported anti-SARS-CoV-2 gene targets or drugs identified from large-scale chemical or genetic screens. These include CRISPR-Cas9 genetic screens in Vero E6 cells (Wei *et al*, 2021) and in cells with exogenous *ACE2* expression (A549$^{ACE2}$; Daniloski *et al*, 2021), and additional lists of experimentally validated drugs reported in different *in vitro* studies compiled by Kuleshov *et al*, 2020 (Materials and Methods). We first tested for significant overlap between our top 10% MTA-predicted targets from each of the datasets and the validation sets described above with Fisher's exact tests (after

mapping all validated target genes or drugs to the metabolic reactions; Materials and Methods). Strongly significant overlaps were found between our predictions from 9 out of the 12 datasets with the antiviral hits (i.e., those whose KO inhibits SARS-CoV-2 infection) identified in the CRISPR-Cas9 screens (all nine cases have FDR < 3.18e-3, the other three have FDR > 0.1; Fig 3B; Table EV6B), these significant datasets include the Vero (Riva *et al*, 2020) and A549$^{ACE2}$ data (Blanco-Melo *et al*, 2020) from the same cell types as those used in the CRISPR-Cas9 screens, but encouragingly also include four *in vivo* patient datasets (Liao *et al*, 2020; Lieberman *et al*, 2020; Xiong *et al*, 2020b; Butler *et al*, 2021). Further examining the experimentally validated anti-SARS-CoV-2 drug sets from previous studies (compiled by Kuleshov *et al*, 2020), we also found a few cases of significant overlap (FDR < 0.1; Fig 3C; Table EV6C). Most of these drug sets are relatively small, but when we pooled all validated drugs compiled by Kuleshov *et al*, their targets are also enriched in the predictions from the BALF, Vero, and 293T datasets (Fig 3C). The top predicted reactions from some datasets are also enriched for host proteins identified to interact with SARS-CoV-2 proteins from Gordon *et al* (2020) and preprint: Stukalov *et al* (2020) (FDR < 0.1; Fig 3D; Table EV6D). Overall, our MTA-based top predictions obtained strong validation from the published CRISPR-Cas9 screens, with additional support from the drug screens and host–virus protein–protein interaction (PPI) data.

We further take advantage of the genome-wide CRISPR-Cas9 screens to more closely evaluate the performance of our MTA predictions. Unlike in many of the drug-screen datasets where the screens are of low-throughput or complete screen results were not available, we were able to confidently define positive and negative sets (i.e., genes whose KO inhibits or promotes the viral infection, respectively) from the CRISPR-Cas9 screen data. The positive and negative sets were defined in a balanced way (Materials and Methods), with which we performed ROC curve analysis of our MTA predictions from each of the datasets (Materials and Methods). Although the MTA prediction is only based on the transformation of cellular metabolic states and does not consider the possible effect of other anti-/pro-viral mechanisms, we see that the predictions based on 6 of the datasets achieved area under ROC curve (AUROC) values above 0.6 and as high as 0.72, although two of the other datasets apparently yielded AUROC significantly lesser than 0.5 (Fig 3E; see

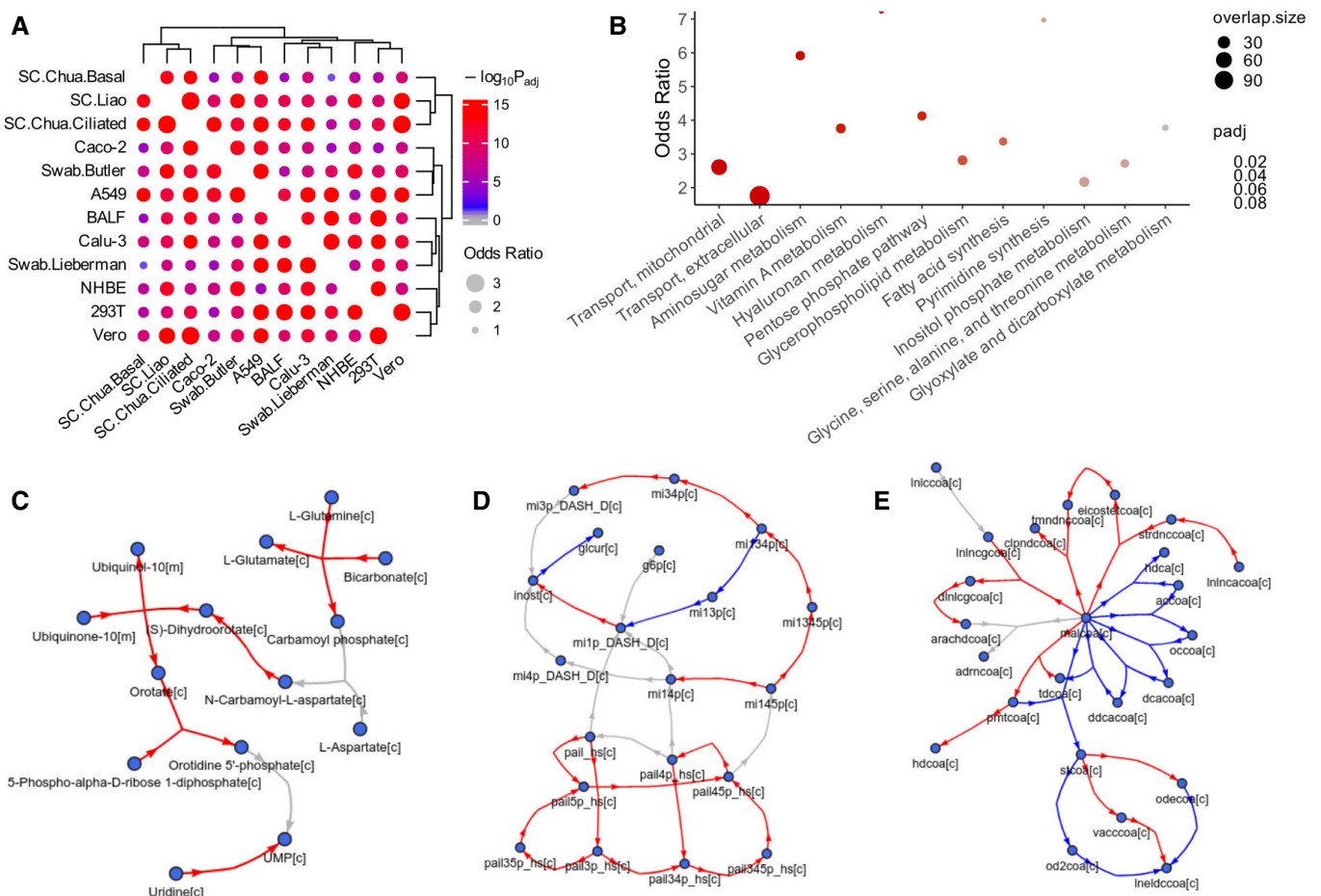

**Figure 2. Genome-scale metabolic modeling (GEM)-based analysis of SARS-CoV-2-induced metabolic alterations across datasets.**

Genome-scale metabolic modeling (GEM) was used to compute the metabolic fluxes from the gene expression profiles, and reactions with differential fluxes (DF) between the SARS-CoV-2-infected and control groups were identified for each dataset (Materials and Methods).

A   Visualization of the overlap of the top DF reactions between each pair of datasets analyzed using Fisher's exact tests (Materials and Methods). The dot size corresponds to the effect size of the overlap as measured by odds ratio, and the color corresponds to the negative $\log_{10}$ adjusted one-sided $P$ value (gray means below 0.05).

B   A summary visualization of the metabolic pathway enrichment result for the top consistent DF reactions across the datasets, with more importance given to the various *in vivo* patient datasets (Materials and Methods). *Y*-axis represents the odds ratio of enrichment, the dot color corresponds to the adjusted $P$ value from Fisher's exact tests, and dot size corresponds to the number of enriched reactions within each pathway. Half-dots plotted on the top border line correspond to infinity odds ratio values. The pathways on the *X*-axis are ordered by $P$ value, and only those with FDR < 0.1 are shown.

C–E   Visualization of the relatively consistent DF patterns in selected enriched pathways. The DF results are based on metabolic modeling using the human GEM Recon 3D (Brunk *et al*, 2018), but for clear visualization, the metabolic network graphs are based on the human GEM Recon 1 (Duarte *et al*, 2007) to reduce the number of metabolites and reactions displayed (Materials and Methods). Metabolites are represented by nodes, reactions are represented by directed (hyper) edges, with edge direction corresponding to the consensus reaction direction and edge color corresponding to the consensus DF direction across datasets (Materials and Methods). Red and blue colors correspond to increased and decreased fluxes, respectively; gray color corresponds to reactions not showing consistent DF changes across datasets, some of such reactions are not shown to increase clarity. (C) Pyrimidine synthesis. (D) Inositol phosphate metabolism. (E) Fatty acid synthesis. Metabolites are labeled by their names in (C) or IDs in (D, E), with suffixes denoting their cellular compartments: [c] cytosol; [m] mitochondria. The mapping between the IDs and metabolite names in (D, E) is given in Table EV5C.

Discussion). As examples, ROC curves from Vero and SC.Liao are shown in Fig 3F, representing the best-performing *in vitro* and *in vivo* datasets, respectively. These results testify that our metabolism-targeting strategy using the MTA algorithm is able to achieve reasonable prediction performances.

Next, we seek to integrate our predictions from the 12 datasets into a final consensus list of high-confidence candidate targets for further extensive experimental validation and investigation. We

applied a procedure to pick highly recurrent top predictions across both the *in vitro* and *in vivo* datasets, as well as from both the Recon 3D- and Recon 1-based results (Materials and Methods), resulting in a final list of 36 candidate target metabolic reactions mapped to 81 genes, and 14 are targeted by known drugs (Table EV7A). This final list of candidates is also strongly enriched for the positive targets identified in the two anti-SARS-CoV-2 CRISPR-Cas9 screens described above (Wei *et al*, 2021 and Daniloski *et al*, 2021; odds

ratio = 8.90, *P* = 1.3e-4). These candidates are enriched for metabolic pathways including cellular transport and inositol phosphate metabolism, among others (FDR < 0.1; Fig 3G; Table EV7B; Materials and Methods). These are consistent with the known biology of SARS-CoV-2; e.g., phosphoinositides are known to be critical for SARS-CoV-2 cell entry by endocytosis, and inhibiting phosphatidylinositol-3,5-bisphosphate with the drug apilimod has been shown to suppress SARS-CoV-2 entry (Ou *et al*, 2020).

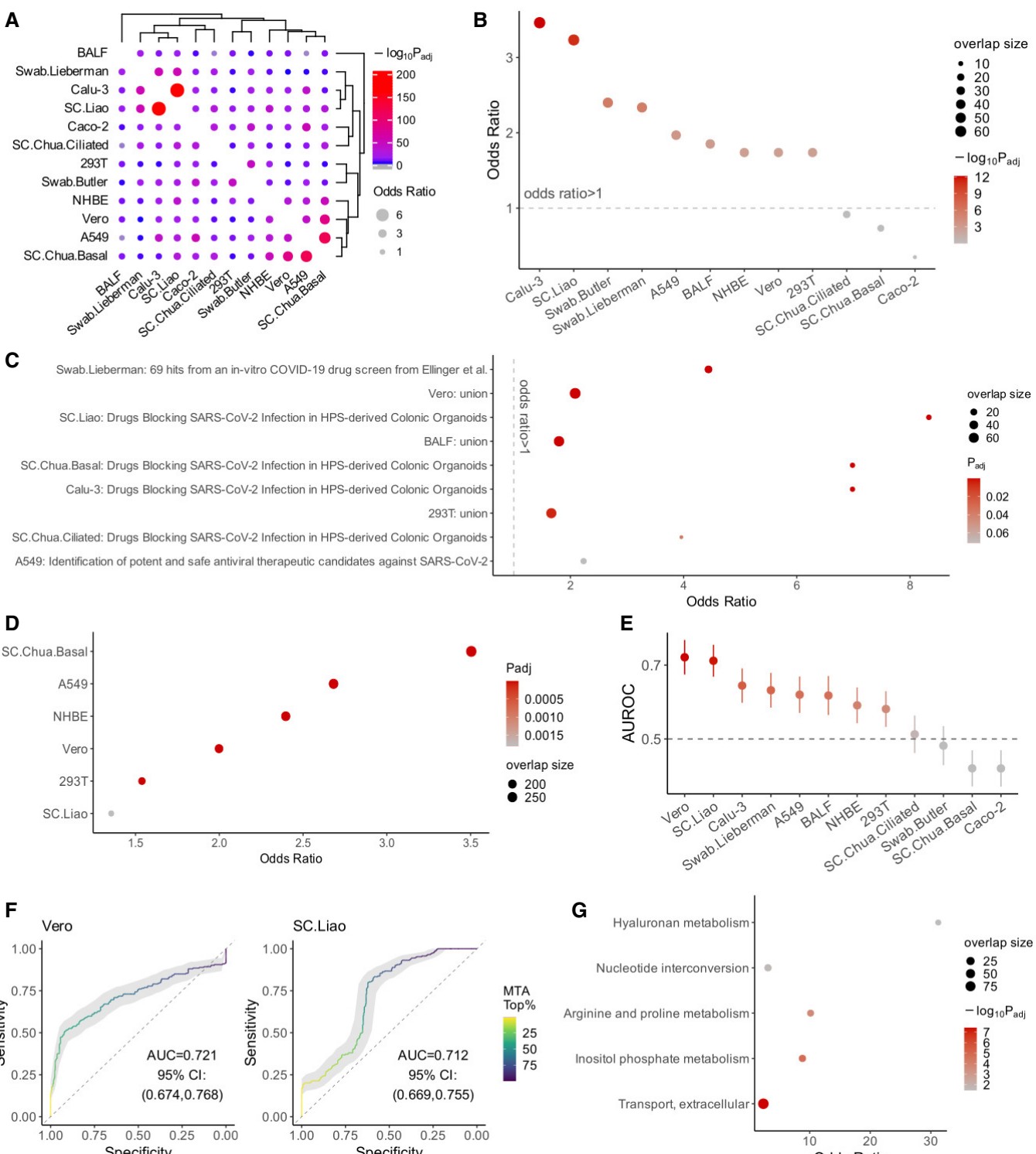

**Figure 3.**

**Figure 3. Genome-scale metabolic modeling (GEM)-based prediction of anti-SARS-CoV-2 targets that act via reversing the virus-induced metabolic alterations.**

The robust metabolic transformation algorithm (rMTA, Valcárcel *et al*, 2019) was used to predict metabolic reactions whose knockout can reverse the SARS-CoV-2-induced metabolic changes using each of the collected datasets (Materials and Methods).

A   Visualization of the overlap of the top 10% MTA-predicted target reactions between each pair of datasets analyzed using Fisher's exact tests (Materials and Methods). The dot size corresponds to the effect size of the overlap as measured by odds ratio, and the color corresponds to the negative $\log_{10}$ adjusted one-sided *P* value (gray means below 0.05).

B   A summary visualization of the enrichment of the top 10% MTA-predicted targets from each dataset for the antiviral hits (i.e., those whose KO inhibits SARS-CoV-2 infection) identified in the two published CRISPR-Cas9 screens (Wei *et al*, 2021 and Daniloski *et al*, 2021; Materials and Methods). *Y*-axis represents the odds ratio of enrichment, the dot color corresponds to the negative $\log_{10}$ adjusted one-sided *P* value from Fisher's exact tests, and dot size corresponds to the number of enriched target reactions. The datasets are ordered by *P* values, the first eight datasets have FDR < 0.1.

C   Cases of significant enrichment (FDR < 0.1) of top 10% MTA-predicted targets from each dataset for the experimentally validated anti-SARS-CoV-2 drug sets from previous studies (compiled by Kuleshov *et al*, 2020). "Union" means the union of all drug sets. Axes and the meanings of dot color and size are similar to (B), but the axes are flipped and the adjusted *P* values are not log-transformed.

D   Significant cases of significant enrichment (FDR < 0.1) of top 10% MTA-predicted targets from each dataset for the host proteins involved in host-SARS-CoV-2 protein–protein interactions (combined from preprint: Stukalov *et al*, 2020 and Gordon *et al*, 2020). Axes and the meanings of dot color and size are similar to (C).

E   A summary of the area under ROC curve (AUROC) value of the MTA prediction using each dataset, based on positive and negative sets (i.e., genes whose KO inhibits or promotes the viral infection, respectively) from the two published CRISPR-Cas9 screen data (Wei *et al*, 2021 and Daniloski *et al*, 2021; Materials and Methods). The error bars (vertical lines through the dots) represent 95% confidence intervals obtained with bootstrapping. The horizontal dashed line corresponds to AUROC of 0.5 (i.e., performance of a random predictor).

F   Example ROC curves from two of the best-performing datasets, one *in vitro* (Vero) and one *in vivo* (SC.Liao). The curves are colored by color gradients corresponding to the threshold of top MTA predictions.

G   Summary of the metabolic pathways significantly enriched (FDR < 0.1) by the final list of consensus candidate targets identified across datasets (Materials and Methods). The axes and the meanings of dot color and size are similar to (B) except that the axes are flipped.

## Validation of the predicted anti-SARS-CoV-2 targets with an *in vitro* siRNA assay

We next seek to experimentally validate the consensus predictions using an *in vitro* siRNA-based target knockdown assay. We have previously conducted a genome-wide siRNA screen to identify host factors that are essential for SARS-CoV-2 replication (Materials and Methods). To further prioritize targets for validation, we applied our computational predictions to this dataset and selected a small subset of 39 genes among the 81 consensus predicted targets (Materials and Methods; target list given in Table EV7C). siRNAs targeting each of these 39 genes were individually transfected into Caco-2 cells ($n = 4$), which were then infected with SARS-CoV-2. Viral replication at 48 hours post-infection was assayed with immunofluorescence labeling of the viral nucleoprotein (*N*) protein, and siRNA-mediated toxicity was evaluated with DAPI staining (illustrated in Fig 4A; Materials and Methods). Overall, compared with negative control non-targeting siRNAs (scrambled), we observed that siRNAs targeting the 39 consensus targets significantly reduced viral replication (Wilcoxon rank-sum test, $P = 1.7e-9$, Fig 4B; raw data in Table EV7D; Materials and Methods). Inspected individually, knocking down 34 of the 39 targets significantly reduced SARS-CoV-2 replication (adjusted $P < 0.05$, Table EV7E). Notably, we also evaluated 4 randomly selected metabolic genes that were not predicted to revert SARS-CoV-2-induced metabolic changes using our analyses (negative controls), which showed much weaker viral inhibition effects (blue dots in Fig 4B). *ACE2* and *TMPRSS2*, two genes known to be essential for SARS-CoV-2 cellular entry (Hoffmann *et al*, 2020), were included as positive controls for comparison (red dots in Fig 4B). Overall, knockdown of the predicted consensus targets did not significantly reduce cell number ($P = 0.73$, Fig 4C), indicating that their impact on viral replication is likely not due to siRNA-mediated cytotoxic effects. Representative fluorescence microscopy images for the siRNA targeting the top three predicted targets (together with the scrambled non-targeting

control and *ACE2* as positive control) are shown in Fig 4D. These results experimentally validate the efficacy of our predicted targets *in vitro*.

## Prediction of metabolic targets for anti-SARS-CoV-2 in combination with remdesivir

Given that our MTA-based prediction of single anti-SARS-CoV-2 metabolic targets has yielded promising results, we proceed to extend the same strategy for the prediction of targets that can be combined with remdesivir to achieve higher antiviral efficacy. To this aim, we cultured Vero E6 cells infected by SARS-CoV-2, with or without remdesivir treatment. A control group (no viral infection or remdesivir treatment) and a remdesivir-only group (no viral infection) were also included (Materials and Methods). Bulk RNA-seq was performed to obtain the gene expression profiles of these samples (Materials and Methods). Visualizing the gene expression data with a PCA plot, we see that remdesivir can indeed effectively reverse the virus-associated expression changes (mostly along the first PC axis), but also results in additional orthogonal changes along the second PC axis (Fig 5A). Performing a GSEA comparing the virus+ remdesivir group with the normal control group, we see that many pathways show significant differences in their expression, including some metabolic pathways, e.g., cholesterol and steroid biosynthesis (Fig 5B; Table EV8; Materials and Methods). Some of these differences can be attributed to the incomplete reversion of virus-induced expression changes by remdesivir, while others may arise from remdesivir-specific effects (Fig 5B). Further computing the metabolic flux profiles representative of each group of samples with iMAT (Shlomi *et al*, 2008) then inspecting the flux-level PCA plot (Fig 5C; Materials and Methods), we observe a similar pattern from that seen on the gene expression level. The DF between the virus+remdesivir and the control group are enriched for various metabolic pathways (FDR < 0.1; Fig 5D; Table EV9; Materials and Methods), many also have DF comparing virus-

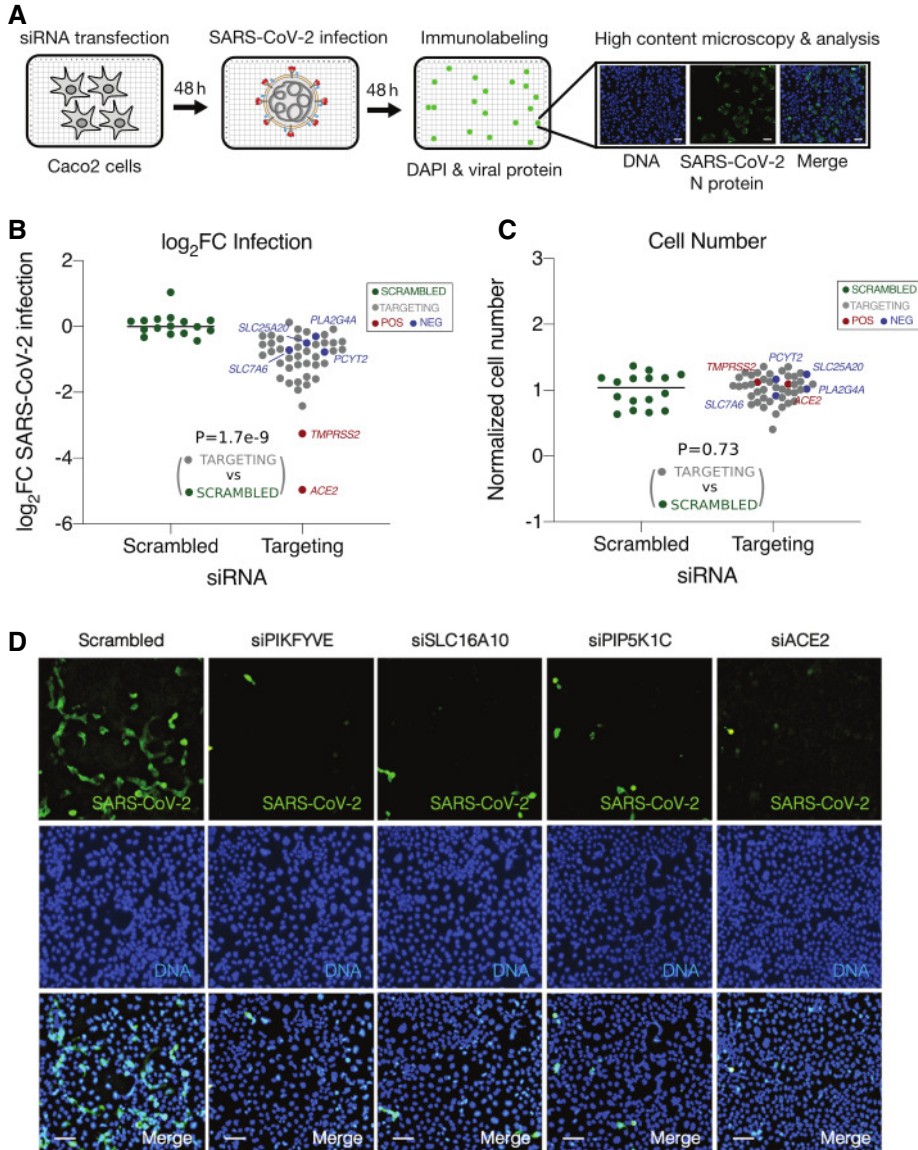

**Figure 4. Validation of the predicted anti-SARS-CoV-2 targets with an immunofluorescence-based *in vitro* siRNA assay.**

A A schematic illustration of the siRNA assay in Caco-2 cells infected with SARS-CoV-2 to validate the antiviral efficacies of the consensus predicted metabolic targets. Caco-2 cells were transfected with siRNAs for 48 h prior to infection with SARS-CoV-2 (MOI = 0.1). Four replicates were performed for each target. At 48 h post-infection, cells were subjected to staining with SARS-CoV-2 nucleoprotein (N) antibody and DAPI, and then imaged to determine the percentage of infected cells after each target knockdown (Materials and Methods).

B Quantification of SARS-CoV-2 infection. After the siRNA knockdown of each target, viral infection was quantified as mean $\log_2$ fold change ($\log_2$FC) of the percentage of SARS-CoV-2$^+$ cells relative to the mean of scrambled non-targeting siRNAs ("SCRAMBLED", green dots; accordingly, the mean $\log_2$FC value of scrambled non-targeting siRNAs was normalized to zero). Predicted positive targets ("TARGETING", gray dots), predicted negative targets ("NEG", blue dots), and positive controls ("POS", red dots, including ACE2 and TMPRSS2) are all shown. Wilcoxon rank-sum test $P$ value comparing the predicted positive targets with scrambled non-targeting siRNAs is given.

C Quantification of cell number after each target knockdown, calculated as the mean fraction of DAPI$^+$ cells relative to the scrambled non-targeting siRNAs (i.e., the latter was normalized to 1). Colors of dots and $P$ value are interpreted in the same way as in (B).

D Representative fluorescence images from the siRNA assay showing SARS-CoV-2 infection (green channel, top row), cell number (blue channel, middle row), and merged cells (bottom row). Results for scrambled non-targeting siRNA as negative control (left column), knockdown of three predicted top metabolic targets (PIKFYVE, SLC16A10, and PIP5K1C, middle columns), and knockdown of positive control (ACE2, right column) are shown. Scale bar = 10 μm.

infected samples with control (see Fig 2B), suggesting that these metabolic changes are not fully reversed to normal by remdesivir. We hypothesize that further reversing the cellular state in the virus+remdesivir group toward the healthy control state may be an effective combinatory targeting strategy to improve the antiviral efficacy of remdesivir.

As before, we focused on the domain of cell metabolism and applied rMTA on our data from Vero E6 cells to predict targets for reversing the metabolic flux profile in the virus+remdesivir group toward normal using both Recon 3D and Recon 1 models (Materials and Methods; Table EV10A). Trying to validate these predictions, we obtained a list of 20 experimentally tested drugs showing synergistic anti-SARS-CoV-2 effects with remdesivir in the Calu-3 cell line (preprint: Nguyenla *et al*, 2020). Despite the cell type difference, we

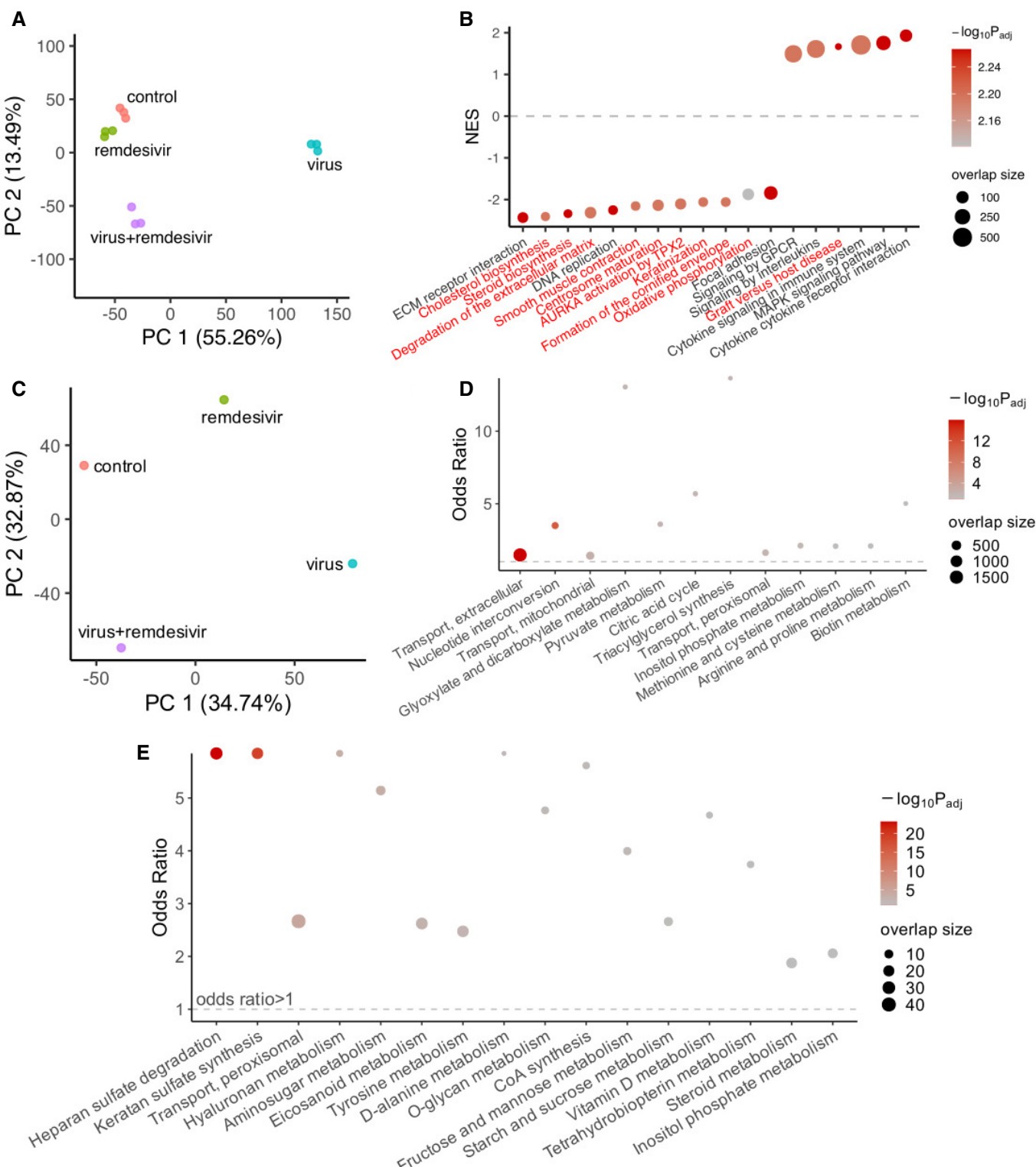

Figure 5.

**Figure 5.  Analysis of the gene expression and metabolic flux profile of remdesivir treatment and prediction of metabolic targets for anti-SARS-CoV-2 in combination with remdesivir.**

A   PCA plot of the gene expression profiles for the Vero E6 samples from all experimental groups: control (no virus or remdesivir treatment), virus (SARS-CoV-2-infected), virus+remdesivir (SARS-CoV-2-infected treated by remdesivir), and remdesivir (remdesivir treatment alone without virus). There are three replicates in each group.

B   A visualization of selected differentially expressed pathways comparing the virus+remdesivir group with the control group using gene set enrichment analysis (GSEA). *Y*-axis represents normalized enrichment score (NES), and positive value means higher expression in the virus + remdesivir group compared with control, vice versa. The dashed line corresponds to NES = 0 (i.e., separating positive and negative enrichment). Pathways on the *x*-axis are ordered by their NES values, all pathways displayed have adjusted *P* < 0.05; pathways with names in red are those that are not significantly different (adjusted *P* > 0.2) when comparing the virus group to the control group using GSEA; i.e., changes in these pathways may arise from remdesivir-specific effects. The dot color corresponds to the negative $\log_{10}$ adjusted GSEA *P* value, and dot size corresponds to the number of enriched genes (i.e., "leading edge" genes in GSEA).

C   PCA plot of the average metabolic flux profile computed using the iMAT algorithm (Shlomi *et al*, 2008; Materials and Methods) representative of each of the experimental groups, the labels are the same as (A).

D   A visualization of metabolic pathway enrichment results of the differential metabolic fluxes in the virus + remdesivir group vs the control group, using Fisher's exact tests (Materials and Methods). *Y*-axis represents the odds ratio of enrichment, the horizontal dashed line corresponds to odds ratio of 1. The dot color corresponds to the negative $\log_{10}$ adjusted one-sided *P* value from Fisher's exact tests, and dot size corresponds to the number of enriched target reactions. The datasets are ordered by *P* values, all pathways displayed have FDR < 0.1.

E   The robust metabolic transformation algorithm (rMTA, Valcárcel *et al*, 2019) was used to predict metabolic reactions whose knockout can further transform the metabolic state of the remdesivir-treated SARS-CoV-2-infected cells back to the normal control state, using the Vero E6 cell samples (Materials and Methods). The significant metabolic pathways (FDR < 0.1) enriched by the top 10% MTA-predicted targets are shown. Axes and the meanings of dot color and size are similar to (D).

observed that the targets of these drugs are significantly enriched by our top 20% MTA predictions when using Recon 1 as the human metabolic model (Fisher's exact test *P* = 0.011, odds ratio 4.83; there is also a trend of enrichment by the top 10% MTA predictions with odds ratio 2.01, but it failed to achieve statistical significance at *P* = 0.30; the enrichment was not significant when using the Recon 3D model). The top 20% Recon 1-based predictions recovered six of the 11 metabolic reaction targets from Nguyenla (preprint: Nguyenla *et al*, 2020), corresponding to drugs including cilostazol, ezetimibe, ivosidenib, and valdecoxib. Some of the top predicted targets overlap with our predicted single anti-SARS-CoV-2 targets as described above, e.g., various inositol phosphate metabolism reactions. This is consistent with the observation that the virus-induced metabolic changes in these pathways are not effectively reversed to normal by remdesivir (as seen from Figs 2B and 5D). Performing a pathway enrichment analysis, we see that the top predictions from both Recon 1 and Recon 3D are enriched in heparan sulfate and hyaluronan metabolism pathways, among others (Fig 5E; Table EV10B); steroid metabolism pathway, which was seen to be different in the gene expression level between the virus + remdesivir and control samples (see Fig 5B), is also enriched for the top predicted targets (Fig 5E). We provide a list of 87 consensus gene targets common to the top 20% predictions from both Recon 1 and Recon 3D (Table EV10A; Materials and Methods). These predictions represent candidate targets that can potentially improve the antiviral efficacy of remdesivir in combination and warrant further testing in future studies.

## Discussion

In this study, we provide a comprehensive GEM analysis integrating 12 published gene expression datasets on SARS-CoV-2 infection, spanning multiple *in vitro* and *in vivo* sample types and expression profiling platforms. We revealed the complexity of host metabolic reprogramming by SARS-CoV-2, and further predicted anti-SARS-CoV-2 single or combinatory (with remdesivir) targets that act via counteracting the virus-induced metabolic changes. Our GEM-based prediction algorithm showed good performance based on validation with published

targets from *in vitro* screens, and the predicted targets represent highly promising candidates for further experimental testing.

To date, a large number of studies have been published that contributed to our fast understanding of the host molecular changes associated with SARS-CoV-2 infection. These studies involve a variety of different experimental models and/or sample types, making it necessary to perform a systematic analysis across datasets and evaluate the robustness and clinical relevance of the findings in human patients. Although we do not aim to (and cannot) include all relevant published data, we tried to cover datasets on both popular *in vitro* models of SARS-CoV-2 infection and human patients (nasopharyngeal swab and BALF samples), aiming to increase the robustness and clinical relevance of our findings. This strategy may also facilitate the testing of our predicted targets and bridge preclinical and potential future clinical drug development. While many other studies have shed light on the systemic and immune cell-specific response characteristic of SARS-CoV-2 infection (e.g., Zheng *et al*, 2020 and many of the patient studies we collected in Table 1), our focus is specifically on the virus-infected host cells, i.e., primarily the airway epithelial cells *in vivo*. Therefore, for human samples, in addition to bulk RNA-seq, we analyzed scRNA-seq data to separate the distinctive changes within the epithelial cells from, e.g., various types of immune cells. In terms of methodology, our complex collection of data from a wide range of platforms with large technical variations (bulk RNA-seq, scRNA-seq, MS-based proteomics) poses a challenge to a formal effect size-based meta-analysis. Despite the progress in multi-omic data integration (Pierre-Jean *et al*, 2020), to the best of our knowledge there is currently no method specifically for integrating the data types we used in this study that are also compatible with our downstream metabolic modeling. Therefore, we instead relied mostly on *P* values, and made subjective decisions that give higher importance to the various patient datasets when defining consistent findings, aiming to obtain results of higher clinical relevance. By integrated analysis of all these data, we found that metabolism is one of the cellular domains that exhibit the most coherent changes across datasets in SARS-CoV-2 infection (besides immune responses; Fig 1D). This finding is consistent with our prior knowledge on the need of a wide spectrum of viruses to manipulate host metabolism for viral proliferation (Mayer

*et al*, 2019). Specific findings from the pathway analysis are also consistent with previous reports; e.g., both TCA cycle and OXPHOS have been shown to decrease based on gene expression during the virus infection (preprint: Ehrlich *et al*, 2020; Gardinassi *et al*, 2020), and have been implicated in the systemic syndromes of the virus (Li *et al*, 2021). Factors involved in sphingolipid metabolism have been found to inhibit the replication of SARS-CoV-2 (Martin-Sancho *et al*, 2021). These set a solid basis for our GEM-based metabolic flux analysis and antiviral target prediction.

The application of GEM in complement to gene expression-level analysis is a central part of our study. It is known that gene expression does not always perfectly correlate with protein level or enzyme activity (Maier *et al*, 2009). Besides, many metabolic reactions are reversible, while the directions of reactions are important biologically, such information is missing on the gene level. By taking advantage of the additional information in the topological constraints of the metabolic network, GEM allows us to infer the actual metabolic fluxes, thus revealing extra complexity in SARS-CoV-2-induced metabolic reprogramming, as is evident from Fig 2C–E. Many of these inferred metabolic changes are consistent with what is known about SARS-CoV-2 and other related viruses. For example, the highly coherent increase in pyrimidine biosynthesis (Fig 2C) corresponds to the increased need of viral genome replication and gene expression (Bojkova *et al*, 2020b), and pyrimidine *de novo* synthesis inhibitors have been shown to have anti-SARS-CoV-2 effects (Xiong *et al*, 2020a). Inositol phosphate metabolism (Fig 2D) is important for the life cycle of many viruses due to the structural or signaling roles of different phosphoinositides (Beziau *et al*, 2020), with the inhibition of certain phosphoinositides disrupting endocytosis and blocking SARS-CoV-2 cell entry (Ou *et al*, 2020). Fatty acid synthesis was reported to increase in SARS-CoV-2 infection (preprint: Ehrlich *et al*, 2020), whereas our results suggest a more complex pattern for different fatty acid species (Fig 2E), which echoes the results of several metabolomics studies (Barberis *et al*, 2020; Shen *et al*, 2020; Thomas *et al*, 2020). Despite that GEM can help to suggest such intricate flux-level patterns, these computed fluxes should be verified with isotope labeling experiments, and their biological significance in the virus infection needs to be further investigated.

Given the importance of metabolism during virus infection, targeting host metabolism has already been proposed as a promising novel antiviral strategy (Mayer *et al*, 2019). For this, the MTA algorithm we previously developed (Yizhak *et al*, 2013), again a method under the GEM framework, can be particularly valuable for the metabolic target discovery. MTA has been successfully applied to predict lifespan extending interventions in yeast (Yizhak *et al*, 2013), a metabolic cancer driver gene (Auslander *et al*, 2017), and a novel therapeutic target for intractable epilepsy (Styr *et al*, 2019). A recent variant of MTA named rMTA was shown to deliver better performance (Valcárcel *et al*, 2019), and here, we used an optimized implementation of rMTA in our study. MTA/rMTA are not based on supervised machine learning techniques and do not use any of the validation datasets for target prediction. Yet, we were able to achieve decent performance during the validation (Fig 3B–F). It is particularly encouraging to see that in several cases, the validation data (which mostly originated from *in vitro* experiments) correlated well with the predictions based on *in vivo* human data. In some datasets, however, our predictions were not successfully validated by the genetic screen data (Fig 2B and E). One reason could be that MTA can only consider the metabolism-related effects and ignores other potential mechanisms that determine the antiviral efficacy of a target. Nevertheless, it could also be due to biological differences between the datasets used for prediction and those used for validation. To avoid overdependence on the limited validation sets available in defining the final consensus candidate target list, we did not explicitly exclude any dataset used for prediction but enforced the inclusion of the human patient data to achieve higher clinical relevance. With such an approach, we were able to identify a set of high-confidence consensus targets, which were then successfully validated with an immunofluorescence-based siRNA assay using SARS-CoV-2-infected Caco-2 cell line. Our assay has the advantage of directly measuring the number of SARS-CoV-2-infected cells via staining of its N protein, while monitoring cytotoxicity (reduction in cell number) via DAPI staining. Reassuringly, knocking down the predicted targets did not exhibit any significant cytotoxicity. These *in vitro* findings should be further validated *in vivo* in future studies. The prediction for combinatory targets with remdesivir also showed promising preliminary results, although our validation is more limited in this case. Our GEM-based pipeline thus complements other computational methods in predicting anti-SARS-CoV-2 targets and drugs, including those based on network analysis (Zhou *et al*, 2020b) or artificial intelligence (Zhou *et al*, 2020c). Follow-up studies are warranted to solidly test and validate these predicted targets for potential further antiviral therapy development.

This study has several limitations that should be more thoroughly addressed in future studies incorporating the GEM modeling approach. First, as we aimed to identify targets in airway epithelial cells, we did not fully characterize cell type- and tissue-specific metabolism associated with SARS-CoV-2 infection. Notably, future studies should analyze single-cell datasets to construct cell type-specific GEMs to identify cell type-specific antiviral targets and virus-induced alterations, e.g., immunometabolic changes (O'Carroll & O'Neill, 2021). Second, it is known that people of different sexes and ages respond differently to SARS-CoV-2 infection (Peckham *et al*, 2020; Canas *et al*, 2021) and that the infection can result in a wide spectrum of disease severity (Sandoval *et al*, 2021), which may be associated with metabolic underpinnings that can in turn be studied in the future with sex, age, and clinical outcome-specific GEMs, given sufficient pertaining preclinical and clinical data. Third, while we predicted combinations of targets based on the principle of restoring cellular homeostasis, it is also feasible to predict synergistic target combinations under the GEM framework via modeling of synthetic lethality, which has also been proposed as a viable antiviral approach (Mast *et al*, 2020).

In summary, we identified prevalent and intricate metabolic reprogramming in the host cell as a feature of SARS-CoV-2 infection, and further predicted single and combinatory antiviral targets with promising performance seen in preliminary validations. These targets should be rigorously validated experimentally. Since our predictions are in part based on human patient data, they are likely to have high clinical relevance and may ultimately help to achieve better efficacy in COVID-19 treatment. Our study demonstrates the targeting of host metabolism as a promising antiviral strategy and highlights the power of GEM analysis to advance the understanding of cell metabolism during viral infection and antiviral target prediction.

# Materials and Methods

### Differential gene expression analysis

We obtained each of the gene expression datasets on SARS-CoV-2 infection from the sources listed in Table 1. For the bulk RNA-sequencing (RNA-seq) datasets whose read count data are available at the time of analysis, we performed differential expression (DE) analysis comparing the SARS-CoV-2-infected or positive samples with the non-infected control or negative samples with DESeq2 (Love *et al*, 2014). For Butler *et al* (2021) and Xiong *et al* (2020b), we obtained the DE results provided from the supplementary materials of the respective publication. For Butler *et al* (2021), among their multiple versions of DE results we used the one from limma-voom with sva correction "Voom:Positive_vs_Negative:10M_samples:sva_correction_2sv". To test whether the mixed use of multiple DE methods could introduce bias to the results, we also used limma-voom (Law *et al*, 2014) on all of the bulk RNA-seq datasets (for results in Appendix Fig S2) and found that the major conclusions were not affected by the change of DE methods. We also took the DE results of the proteomic data from Bojkova *et al* (2020b) as provided by the authors, and used the 24-h post-infection data, which is the latest time point available with the largest number of DE proteins. For the single-cell RNA-sequencing (scRNA-seq) datasets, The "FindMarkers" function in the R package Seurat (Stuart *et al*, 2019) was used to call the MAST method (Finak *et al*, 2015) for DE analysis in each annotated cell type, with "logfc.threshold" set to 0 to obtain full results across genes. We focused on the airway epithelial cells since our major aim in this study is to investigate the changes in the cell types infected by the SARS-CoV-2 virus, these include the "Epithelial" cell type from Liao *et al*, 2020 and the "Ciliated" and "Basal" cell types from Chua *et al*, 2020 (Data ref: Chua *et al*, 2020; other epithelial subtypes from these datasets yielded no significant DE genes). All DE results are given in Table EV1.

### Gene set enrichment analysis of the differential expression results

Using the DE log fold change values from each dataset, gene set enrichment analysis (GSEA) (Subramanian *et al*, 2005) was performed using the implementation in the R package fgsea (preprint: Korotkevich *et al*, 2021). The gene set/pathway annotations used were the Reactome (Jassal *et al*, 2020) and KEGG (Kanehisa *et al*, 2021) subsets from the "Canonical Pathway" category in version 7.0 MSigDB database (Liberzon *et al*, 2011). For metabolic pathways (in Fig 1E), those under the category "Metabolism" from KEGG (Kanehisa *et al*, 2021) were used. All GSEA results are given in Table EV2.

### Comparison of the differentially expressed genes and pathways across datasets

The DE results across datasets were compared in a descriptive manner. As a first approach, the DE log fold change values were inverse normal-transformed across all genes within each dataset, which preserves only the order (i.e., rank) of DE effect sizes, and then, PCA was applied to the transformed data. As a second approach, top significantly DE genes or enriched pathways with

FDR < 0.1 from each pair of datasets were tested for significant overlap using Fisher's exact tests. To identify the consistent DE changes across datasets, a formal meta-analysis of all 12 datasets is challenging given the wide range of assay platforms and DE algorithms used. So instead, we adopted subjective criteria that give high importance to the various *in vivo* patient datasets, such that the results may be more clinically relevant: We identified pathways that are significantly (FDR < 0.1) enriched in the consistent direction (up/down-regulation) in at least one of the bulk RNA-seq patient datasets and also at least one of the scRNA-seq datasets, while never showing significant enrichment (FDR < 0.1) in the opposite direction in any of the datasets (for the results in Fig 1D; Table EV3).

### Computation of metabolic fluxes from gene expression data with genome-scale metabolic modeling

For each dataset, we used the genome-scale metabolic modeling (GEM) algorithm iMAT (Shlomi *et al*, 2008) to compute the metabolic flux profile from gene expression data. iMAT requires gene-length-normalized expression values in the bulk RNA-seq datasets; for this, we computed TPM values with Salmon (Patro *et al*, 2017) from the raw fastq files for datasets where TPM data are not provided. Then for each dataset, we took the median expression values of the control and virus-infected samples, respectively, as the representative expression profile for each group, and used it as input to iMAT. The human genome-scale metabolic model (GEM) Recon 3D (Brunk *et al*, 2018) was used as the base model for iMAT. Since Recon 3D is a large model that in some cases may pose difficulty to the mixed integer programming (MIP) solver used in iMAT, we also used an older and smaller version of the human GEM, i.e., Recon 1 (Duarte *et al*, 2007) to double-check the numerical stability and robustness of results. The output of iMAT is a refined GEM for the each of the virus-infected and control groups in each dataset, with metabolic reaction bounds adjusted to achieve maximal concordance with the gene expression data while satisfying the stoichiometric constraints of the cellular metabolic network (Shlomi *et al*, 2008). Each output model defines a space rather than a single unique solution of the global metabolic flux profile, and artificial centering hit-and-run (ACHR) was used to sample the metabolic space and obtain the distribution of flux values for each metabolic reaction in each condition (control or virus-infected) and dataset. Although not a single value representing a unique solution, we were able to determine reactions with DF by comparing the flux distributions of a reaction in control and virus-infected conditions (see next section). We did not apply flux balance analysis (FBA) on the iMAT-derived constrained models, as maximal biomass production may not be appropriate especially for the *in vivo* patient samples, and tissue type-specific objective functions for these samples are not trivial to define. All GEM analyses were performed using our in-house R package named gembox, with the academic version of IBM ILOG CPLEX Optimization Studio 12.10 as the optimization solver on a high-performance computing cluster.

### Differential flux analysis of virus-infected vs control group in each dataset

The flux distributions of the control and infected groups were compared to identify reactions with DF. Since an arbitrarily large

number of sample points can be sampled from the metabolic space of each group, resulting in statistical tests with arbitrarily small $P$ values, we adopted the following effect size-based criterion for DF reactions: absolute rank biserial correlation (an effect size measure of the difference between the two flux distributions in the control and virus-infected groups) > 0.5, and absolute relative flux change (i.e., the absolute difference of the mean fluxes between the two groups over the absolute mean flux in the control group) > 50%. Positive DF reactions have flux value difference in infected vs control group > 0, and vice versa for negative DF reactions. Note that for non-reversible reactions, flux values are non-negative and the sign of DF can be interpreted similar to differential gene expression; for reversible reactions, flux values can be negative, representing reactions happening in the reverse direction; thus, the sign of DF needs to be interpreted differently; e.g., negative DF represents flux shift toward the reverse direction and not necessarily decreases in absolute flux. The DF results are given in Table EV4, these are based on modeling results with Recon 3D.

### Analysis of reactions with consistent differential fluxes across datasets and their pathway enrichment analysis

To compare the DF results across datasets, the DF reactions from each pair of datasets were tested for significant overlap using Fisher's exact tests (separately for positive and negative DF). Since no reaction shows fully consistent DF across all 12 datasets analyzed, similarly as with the DE analysis, we identified the DF reactions with high level of consistency especially in the *in vivo* patient datasets, such that the results may be more clinically relevant: We identified DF reactions in the consistent direction (positive/negative) in at least one of the bulk RNA-seq patient datasets and also at least one of the scRNA-seq datasets, while showing DF in the opposite direction in no more than three datasets (Table EV5A). The metabolic pathway enrichment of these DF reactions was analyzed with Fisher's exact tests (results in Table EV5B), with the "subSystems" slot in the Recon 3D metabolic model used as pathway annotation. However, we also performed the enrichment analysis with the Recon 1 modeling results and the corresponding "subSystems" annotation, and we removed pathways that show inconsistent enrichment results between Recon 3D and Recon 1. Note that due to the special interpretation of the sign of DF values as explained above, the GSEA used for gene expression-level analysis is not appropriate for pathway enrichment analysis on the flux level.

### Analysis of the consistent flux alteration patterns in different metabolic pathways

For each of the significantly enriched metabolic pathways identified in the consistent DF reaction analysis described above, we defined the "consensus" direction of each reaction as represented by those shown in the virus-infected group from the majority (> 6 out of 12) of the datasets, and also similarly for the "consensus" direction of DF for each reaction. The consensus directions of reactions and their DF were overlaid onto network diagrams of the pathways and visualized, where metabolites are represented by nodes, reactions are represented by directed (hyper) edges with edge direction corresponding to the consensus reaction direction and edge color corresponding to the consensus DF direction. Parts of the metabolic pathways where reactions are not consistently altered across datasets are grayed out or removed to increase the clarity. The DF results from the Recon 3D model (Brunk *et al*, 2018) was used, but for clear visualization, the network diagrams of the metabolic pathways are based on the smaller Recon 1 model (Duarte *et al*, 2007) to reduce the number of metabolites and reactions displayed. Common reactions shared by Recon 3D and Recon 1 were mapped by their IDs when the IDs are the same, or were manually mapped according to the metabolite interconversion relationship when the IDs are different. Further, upon visual inspection, potential futile loops in the network are also removed from the visualizations.

### Prediction of anti-SARS-CoV-2 target metabolic reactions with metabolic transformation algorithm

For each of the collected datasets, the DE result of virus-infected vs control samples and the representative flux distribution of the virus-infected group computed with iMAT (Shlomi *et al*, 2008) followed by ACHR sampling were used as inputs for the GEM-based MTA (Yizhak *et al*, 2013; a variant called rMTA was used; Valcárcel *et al*, 2019) to predict metabolic reactions whose knockout can transform cellular metabolic state from that of the virus-infected to that of the control samples (full prediction results from all datasets in Table EV6A). The output of rMTA is a score (rMTA score) for each metabolic reaction, with higher scores corresponding to better candidates for achieving the metabolic transformation as specified above. From our previous experience (Yizhak *et al*, 2013), the top 10–20% MTA predictions contain promising targets. The human Recon 3D (Brunk *et al*, 2018) GEM was used for the MTA analysis, and we also used Recon 1 GEM (Duarte *et al*, 2007) to confirm the robust predictions. The rMTA algorithm implemented in our in-house R package named gembox was used, with the academic version of IBM ILOG CPLEX Optimization Studio 12.10 as the optimization solver on a high-performance computing cluster. To compare the MTA predictions across datasets, the top 10% predictions from each pair of datasets were tested for significant overlap using Fisher's exact tests.

### Computational validation of the MTA-predicted anti-SARS-CoV-2 metabolic targets

Multiple datasets of reported anti-SARS-CoV-2 gene targets or drugs identified from large-scale chemical or genetic screens were collected to validate our predictions. Gene-level results of two published CRISPR-Cas9 genetic screens (Wei *et al*, 2021 and Daniloski *et al*, 2021) were obtained from the supplementary materials of the respective publication. For Wei *et al*, gene hits with FDR < 0.1 and mean $z$ score > 0 (i.e., KO inhibits the viral infection) were taken; Daniloski *et al* reported two screens with different multiplicities of infections (MOIs) and provided only single-sided FDR, so gene hits with FDR < 0.1 from either screen were taken. The union set of hits from both studies were used. Lists of experimentally validated drugs reported in different studies compiled by Kuleshov *et al* (2020) were downloaded from https://maayanlab.cloud/covid19/, which are then mapped to the genes they inhibit using data from DrugBank v5.1.7 (Wishart *et al*, 2018). Additionally, host proteins identified to interact with SARS-CoV-2 proteins were obtained from the supplementary materials of Gordon *et al* (2020) and preprint: Stukalov *et al* (2020). The genes from these

validation datasets are mapped to metabolic reactions wherever applicable based on the human GEM Recon 3D (Brunk *et al*, 2018) data. Then, the significant overlap between the top 10% MTA-predicted targets from each dataset and each of the validation sets described above was tested with Fisher's exact tests on the reaction level (full results in Table EV6B–D). Reaction-level test is performed because multiple reactions can be mapped to the same gene, and performing Fisher's exact test on the gene level fails to consider such multiple mapping and is thus inappropriate.

For ROC analysis, negative sets (i.e., genes whose KO promotes SARS-CoV-2 infection) were defined based on the two CRISPR-Cas9 screens described above. For Wei *et al*, gene with FDR < 0.1 and mean $z$ score < 0 were taken; since Daniloski *et al* provided only single-sided FDR, the log fold change threshold corresponding to the FDR < 0.1 cutoff was identified, and genes with more extreme log fold changes in the opposite direction were taken. The union of the negative sets from both studies was used. Both the positive (described in the previous paragraph) and negative sets of genes are then mapped to metabolic reactions as described above. The negative set defined as such contains a relatively balanced number of reactions compared with the positive set (306 vs 238). The rMTA score for the reactions produced by MTA was used as the predicted value for ROC analysis. The R package pROC (Robin *et al*, 2011) was used to compute the AUROC values and their 95% confidence intervals (the latter computed with bootstrapping).

### Defining and analyzing the consensus set of candidate anti-SARS-CoV-2 metabolic targets across datasets

Based on top 10% MTA predictions from the 12 datasets (six *in vitro* and six *in vivo*) using Recon 3D, the metabolic reaction targets that are recurrent in at least two of the *in vitro* datasets, and also in two of the *in vivo* datasets (i.e., the intersection of the two) were taken, and were then mapped to genes based on the model data. This procedure was repeated for the Recon 1-based predictions, and the intersection between the Recon 3D predictions and Recon 1 predictions was taken to be the final consensus candidate gene targets with high-confidence support across datasets. These target genes were further mapped to known drugs inhibiting the gene targets using data from DrugBank v5.1.7 (Wishart *et al*, 2018; target list given in Table EV7A; the reaction and rMTA score information in this table was based on the Recon 3D results). The metabolic pathway enrichment of these targets was analyzed with Fisher's exact tests, with the "subSystems" slot in the metabolic model used as pathway annotation. For our siRNA assay-based experimental validation of the predictions, we focused on a further subset of those consensus candidate gene targets with negative log fold change regardless of $P$ value in a previous genome-wide siRNA screen we performed (data deposited at https://figshare.com/s/4117ac39b1d21b56f5e6); namely, our computational predictions were used to prioritize the targets for focused replicated validation assays from the much more noisy results of genome-wide screens. This list of targets for experimental validation is given in Table EV7C.

### Validation of the consensus set of predicted anti-SARS-CoV-2 targets with siRNA assay

A targeted small-scale siRNA screen was carried out in human Caco-2 cells to evaluate whether the predicted metabolic targets affect the replication of SARS-CoV-2. The siRNAs (ON-TARGETplus SMART-pool, Dharmacon) were individually arrayed in 384-well plates at a concentration of 12.5 nM per well. In addition, non-targeting siRNAs (scrambled) were added to each plate as negative controls, and siRNAs targeting SARS-CoV-2 entry factors *ACE2* and *TMPRSS2* were included as positive controls. siRNAs were mixed with 0.1 μl Lipofectamine RNAiMAX transfection reagent diluted in 9.90 μl Opti-MEM (both reagents from Thermo Fisher Scientific) to enable the formation of siRNA transfection reagent complexes. Following a 20-min incubation period at room temperature, 3,000 Caco-2 cells diluted in 40 μl DMEM (Gibco) supplemented with 10% heat-inactivated fetal bovine serum (FBS, Gibco), and 50 U/ml peni-cillin–50 μg/ml streptomycin (Fisher Scientific) were seeded on top of the complexes and incubated for 48 h at 37°C, 5% $CO_2$. Cells were then infected with SARS-CoV-2 (USA-WA1/2020) at a multi-plicity of infection (MOI = 0.1) for 48 h at 37°C, 5% $CO_2$, and then fixed with 4% PFA (Boston BioProducts) for 4 h at room tempera-ture. Cells were then washed twice with PBS, permeabilized with 0.5% Triton X-100 for 20 min, followed by blocking with 3% BSA (Sigma) for 1 h at room temperature. Primary anti-SARS-CoV-2 N protein rabbit polyclonal antibody (gift from Dr. Adolfo Garcia-Sastre) was added for 2 h at room temperature, followed by three washes with PBS and 1-h incubation with Alexa Fluor 488-conjugated anti-rabbit secondary antibody (Thermo Fisher Scien-tific) diluted in 3% BSA. Following three washes with PBS, cells were stained with DAPI (4,6-diamidine-2-phenylindole, KPL), and plates were sealed and stored at 4°C until imaging. SARS-CoV-2 replication after each individual target knockdown was quantified using high-content imaging. The assay plates were imaged with the IC200 imaging system (Vala Sciences) at the Conrad Prebys Center for Chemical Genomics (CPCCG) and analyzed using the analysis software Columbus v2.5 (PerkinElmer). Based on the number of Alexa 488+ objects and the number of DAPI$^+$ objects, the percentage of infected cells was quantified. The $\log_2$FC infection was calculated relative to the negative control scrambled siRNA-treated wells. Cyto-toxicity resulting from siRNA transfection was evaluated by normal-izing the percentage of DAPI$^+$ objects to that of the negative control scrambled siRNA. All experiments dealing with SARS-CoV-2 were performed in a Biosafety Level 3 laboratory under the approval of the Sanford Burnham Prebys Medical Discovery Institute Biosafety Committee.

### Preparation of Vero E6 cell samples with SARS-CoV-2 infection and remdesivir treatment, RNA-sequencing, and gene expression data analysis

Vero E6 cells (ATCC® CRL-1586™) were maintained in Dulbecco's modified Eagle's medium (DMEM, Gibco) supplemented with 10% heat-inactivated fetal bovine serum (FBS, Gibco), 50 U/ml peni-cillin, 50 μg/ml streptomycin, 1 mM sodium pyruvate (Gibco), 10 mM 4-(2-hydroxyethyl)-1-piperazine ethanesulfonic acid (HEPES, Gibco), and 1× MEM non-essential amino acid solution (Gibco). The SARS-CoV-2 USA-WA1/2020 strain was obtained from BEI Resources (NR-52281). The virus was inoculated on Vero E6 cells, and the cell supernatant was collected at 72 h post-inoculation (hpi), when extensive cytopathic effects were observed. The super-natant, after clarification by centrifugation for 15 min at 4°C at 5,000 *g*, was aliquoted and stored at −80°C until use. 500,000 Vero

E6 cells were seeded in six-well plates. The following day, the cell medium was replaced with fresh medium supplemented with either DMSO or 1 μM remdesivir (Adooq Bioscience), and cells were either mock-infected or infected with SARS-CoV-2 USA-WA1/2020 (MOI = 0.3). Twenty-four hours after infection, cells were collected, and total intracellular RNA was extracted using the Qiagen® RNeasy® Plus Mini Kit. Three replicates were performed for each group, resulting in a total of six samples. The quality of the extracted RNA was assessed with Agilent® 2100 Bioanalyzer. Libraries were prepared on total RNA following ribosome RNA depletion with standard protocol according to Illumina®. Total RNA-sequencing was then performed on the Illumina® NextSeq system, 150-bp paired-end runs were performed, and 100 million raw reads per sample were generated. STAR (Dobin *et al*, 2013) was used to align the reads to reference genome of the African green monkey (*Chlorocebus sabaeus*, https://useast.ensembl.org/Chlorocebus_sabaeus/Info/Annotation), with the SARS-CoV-2 genome (https://www.ncbi.nlm.nih.gov/nuccore/NC_045512) added to the reference genome. DESeq2 (Love *et al*, 2014) was used for DE analysis between pairs of experimental groups (including virus+remdesivir vs control and virus vs control; DE results in Table EV8A). GSEA (Subramanian *et al*, 2005) was performed using the implementation in the R package fgsea (preprint: Korotkevich *et al*, 2021), results are provided in Table EV8B. The gene set/pathway annotations used were the Reactome (Jassal *et al*, 2020) and KEGG (Kanehisa *et al*, 2021) subsets from the "Canonical Pathway" category in version 7.0 MSigDB database (Liberzon *et al*, 2011). All experiments dealing with SARS-CoV-2 were performed in a Biosafety Level 3 laboratory under the approval of the Sanford Burnham Prebys Medical Discovery Institute Biosafety Committee.

### Genome-scale metabolic modeling of the remdesivir-treated Vero E6 cell samples and prediction of anti-SARS-CoV-2 metabolic targets in combination with remdesivir

As with the metabolic modeling of the other datasets on SARS-CoV-2 infection, iMAT (Shlomi *et al*, 2008) together with ACHR was used to compute the metabolic flux distribution for each of the experimental groups, using the median expression TPM values of each group as the input to iMAT. Reactions with DF between groups (including virus + remdesivir vs control and virus vs control) were identified as described above, and their significant metabolic pathway enrichment was tested with Fisher's exact tests, with pathways defined by the "subSystems" from the Recon 3D model (Brunk *et al*, 2018; results in Table EV9). Like above, the smaller Recon 1 model (Duarte *et al*, 2007) was also used to identify robust findings, and non-robust pathway-level results were discarded similarly as above and not considered in the main text. The DE result of virus+remdesivir vs control group and the mean flux distribution of the virus+remdesivir group computed with iMAT were used as inputs for rMTA to predict metabolic reactions whose knockout can further transform the virus+remdesivir metabolic state to the normal control state. The top 10% and 20% MTA-predicted targets from either Recon 3D or Recon 1 were tested for significant enrichment for the targets of a list of experimentally validated synergistic drugs with remdesivir (preprint: Nguyenla *et al*, 2020) using Fisher's exact test (performed on the metabolic reaction level as described above). Metabolic pathway

enrichment analysis of the top rMTA-predicted targets was performed as described above (results in Table EV10B). The top 20% reactions in Recon 1- and Recon 3D-based predictions were mapped to genes, and the intersection between the two sets of predicted genes were taken as a final consensus list of candidate gene targets. Like above, these target genes were also further mapped to known drugs inhibiting the gene targets using data from DrugBank v5.1.7 (Wishart *et al*, 2018; target list given in Table EV10A; the reaction and rMTA score information in this table was based on the Recon 3D results).

### Notes on statistical analysis and visualization

R version 3.6.3 was used for all statistical tests. *P* values lesser than 2.22e-16 may not be computed accurately and are reported as "*P* < 2.22e-16" throughout the text. The Benjamini–Hochberg (BH) method was used for *P* value adjustment throughout the text. The R packages ggplot2 (Wickham, 2016), ComplexHeatmap (Gu *et al*, 2016), and visNetwork (https://cran.r-project.org/web/packages/visNetwork/index.html) were used to create the visualizations.

# Data and code availability

The gene expression data analyzed in this study are from published studies, with detailed information given in Table 1. The bulk RNA-seq data for SARS-CoV-2 infection in Vero E6 cells with remdesivir treatment have been deposited to the GEO database (accession ID: GSE165955; http://www.ncbi.nlm.nih.gov/geo/query/acc.cgi?acc=GSE165955). The code used for the analyses can be found in the GitHub repository: https://github.com/ruppinlab/covid_metabolism. Our in-house R package named gembox used for all the GEM analysis in this study can be found on GitHub: https://github.com/ruppinlab/gembox.

## Acknowledgements

This research was supported in part by the Intramural Research Program of the National Institutes of Health, NCI, and used the computational resources of the NIH HPC Biowulf cluster (http://hpc.nih.gov). We acknowledge and thank the National Cancer Institute for providing financial and infrastructural support. The following reagent was deposited by the Centers for Disease Control and Prevention and obtained through BEI Resources, NIAID, NIH: SARS-Related Coronavirus 2, Isolate USA-WA1/2020, NR-52281. This work was also supported by the following grants to the Sanford Burnham Prebys Medical Discovery Institute: DoD: W81XWH-20-1-0270; DHIPC: U19 AI118610; Fluomics/NOSI: U19 AI135972. K.C. and S.S are supported by the NCI-UMD Partnership for Integrative Cancer Research Program.

## Author contributions

KC, SS, and ER conceived the study and designed the computational analyses. KC, SS, LRP, and NUN collected the data. KC developed the software tools and performed the analyses. LR performed the experiments on remdesivir treatment. LM-S and YP performed the siRNA experiments for target validation. LM-S and XY provided the genome-wide siRNA screen data for SARS-CoV-2. SKC and ER provided funding and supervised the study. KC, LR, LM-S, SS, LRP, NUN, and ER wrote the paper. All authors approved the publishing of the manuscript.

## Conflict of interest

The authors declare that they have no conflict of interest.

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
