## [Review Process File · Molecular Systems Biology]

Genome-scale metabolic modeling reveals SARS-CoV-2-induced metabolic changes and antiviral targets

Kuoyuan Cheng, Laura Martin-Sancho, Lipika Pal, Yuan Pu, Laura Riva, Xin Yin, Sanju Sinha, Nishanth Nair, Sumit Chanda, and Eytan Ruppin

DOI: [10.15252/msb.202110260](https://doi.org/10.15252/msb.202110260)

Corresponding author(s): Eytan Ruppin (eyruppin@gmail.com) , Sumit Chanda (schanda@sbpdiscovery.org)

Review Timeline:

Submission Date:	30th Jan 21
Editorial Decision:	28th Feb 21
Revision Received:	24th Aug 21
Editorial Decision:	27th Sep 21
Revision Received:	29th Sep 21
Accepted:	30th Sep 21

Editor: Jingyi Hou

Transaction Report:

Thank you for submitting your work to Molecular Systems Biology. We have now heard back from the three reviewers who agreed to evaluate your manuscript. As you will see from the reports below, the reviewers acknowledge the potential interest of the study. However, they raise a series of concerns, which we would ask you to address in a major revision.

Without reiterating all the points raised in the reviews below, some of the more substantial issues are the following:

- Provide further experimental validations for several of the identified targets to support the conclusions, as suggested by Reviewer #2.
- Reviewers raise concerns about constraints used in the modeling and the use of Recon1 instead of a more updated model, which need to be carefully addressed.

All other issues raised by the reviewers need to be satisfactorily addressed as well. As you may already know, our editorial policy allows in principle a single round of major revision, and it is therefore essential to provide responses to the reviewers' comments that are as complete as possible.

On a more editorial level, we would ask you to address the following issues:

REFEREE REPORTS

Reviewer #1:

Cheng and his colleagues performed genome-scale metabolic modeling observations for SARS-CoV-2 infected host cells to identify possible antiviral targets for COVID-19. Specifically, they built genome-scale metabolic models using 12 publicly available transcriptome data for SARS-CoV-2 infected host cells. These observations reveal new host metabolism reprogramming during SARS-CoV-2 infections and offered a few potential host factors related to SARS-CoV-2 infections, which were validated by published CRISPR-Cas9 screening data. Via analyzing RNA-sequencing data of remdesivir-treated Vero E6 cell samples, they prioritized several metabolic targets in possible combination with remdesivir. Although in silico findings were not validated by experimental assays, this study presented a new in silico methodology for development of COVID-19 therapeutic strategies from host metabolism perspectives. The manuscript is clearly written, the methods are described in sufficient detail and the conclusions are fairly supported by the data. Several specific comments need attention below, which may improve the manuscript further.

1. The authors inspected 12 publicly available transcriptome datasets for SARS-CoV-2 infected host samples. It will be interesting the authors can discuss cell type-specific and tissue-specific genome-scale metabolic models for SARS-CoV-2

infections, which will provide cell type-specific drug targets for effective therapeutic development.

2. The authors identified several interacting metabolic pathways, such as Branched-chain amino acid (valine, leucine). Could the authors discuss possibility of SARS-CoV-2 infection mediated metabolism is sex- or age-specific. For example, branched-chain amino acid pathway is highly age-specific. It will help to better understand the age-specific morbidity in COVID-19 from genome-scale metabolic modelings.

3. Children and older individuals have different responses to SARS-CoV-2 infectious. Individuals with SARS-CoV-2 infectious have different outcomes, including symptomatic vs. asymptomatic, mild, moderate and severe outcomes. It will be interesting the authors can discuss COVID-19 outcome-specific genome-scale metabolic models, which will help us understand disease severities of COVID-19 from metabolism perspectives.

4. Several related literatures may be discussed further, such as doi: 10.1371/journal.pbio.3000970 and doi: 10.1016/S2589-7500(20)30192-8.

5. Overall, this is a novel, computational study from SARS-Cov-2 host metabolism perspective. The authors may discuss future works, about metabolic synthetic lethality and metabolic vulnerability studies, for emerging development of host-targeting therapies for COVID-19.

Reviewer #2:

The authors presented an integrated analysis of 12 published in vitro and human patient gene expression datasets on SARS-CoV-2 infection using genome-scale metabolic modeling (GEM) and revealed the reprogramming of host metabolism during SARS-CoV-2 infection. They also applied the GEM based metabolic transformation algorithm to predict anti-SARS-CoV-2 targets that counteract the virus-induced metabolic changes and validated them using published experimental drug and genetic screen data. Moreover, they generate and analyzed RNA-sequencing data of remdesivir treated Vero E6 cell samples and predicted metabolic targets acting in combination with remdesivir. They provided top high-confidence candidate targets with support from human patient data for evaluation in further studies. This is a very interesting study; however, it lacks experimental evidence for the targets identified in the study. I strongly suggest the authors validate their top predictions by performing additional experiments.

Major Comments

1. Flux balance analysis renders a considerable part of the result and contributes significantly to the main conclusion and discovery in this study. However, there is some missing information about how the modeling is performed and what rationality is. First of all, there is no information about constraints used in the modeling, which is a very important part of flux balance analysis and could significantly affect the final results. It could be that the author used the default constraints from Recon1, but it then raises another very serious concern if the rationale is to use the same constraints to model in vitro cell lines and in vivo human cells. At the very beginning, it may be useful to compare the fluxes between cell lines and human tissues as the rate of the flux is expected to be at a different scale.

2. Recon1 is used for the FBA in this study, and the author claimed this is because using more recent models will significantly increase the computational cost. This is not acceptable since there are many different updated models since 2017. Recon 3 and Human1 can be used. This is also an unfair statement, as the scale between Recon1 and the most recent GEM Recon 3 and Human1 is not significant and the computational cost for running iMAT using these models will not be that different. In addition, Recon1 has many mass imbalanced reactions which could affect the FBA results and lead to incorrect conclusions. So, I would argue that the authors should use one of the most recent human GEM for the FBA analysis.

3. One of the very important contributions of this study is that the author proposed some promising targets that could be used together with remdesivir for COVID19 treatment based on their in-house in vitro experimental data. This means their lab is capable of performing COVID19 in vitro tests. However, I am surprised that they have not tested the effect of the targets (top candidate(s)) they identified using siRNA or plasmid as that should be relatively simple compared to what they have already done. The conclusion of this study will be significantly enhanced if the authors perform this experiment. The readers of the MSB would also expect to see the validation experiments.

4. There are many parameters proposed by the authors arbitrarily selected and need sensitivity analysis. For example, the authors selected 'top 400' DE genes with at least 'FDR <0.1' for all different COVID19 datasets, which seems a very sensitive cutoff. Why 400? Will the result change significantly if you choose top 500? Also, the authors selected 'FDR<0.2' for metabolic pathway meta-analysis, which is neither statistically sound nor consistent with the other cutoff used in this study.

5. The author claimed that they used an in-house R package named 'gembox' for the GEM based analysis. The authors should provide the script for the analysis to enable proper assessment.

Reviewer #3:

Summary

Cheng et al utilizes 12 published SARS-CoV2 gene expression data sets to investigate virus-induced host metabolic changes, using genome scale metabolic modeling. Based on these identified metabolic shifts, they then identified potential metabolic drug targets using MTA, and compared these proposed metabolic targets to published drug and genetic screening data. Lastly, they investigated potential coupled therapeutics through their remdesvir-treated Vero E6 cell RNA-seq data. The goal of this paper is to utilize genome scale metabolic modeling to identify potential host treatment avenues for SARS-CoV2 infection, as opposed to the more common approach of identifying viral targets. This unique approach to viral-infection therapeutic development creates a pipeline for drug target discovery that could be applied to other viral infections outside of SARS-CoV2. Additionally the use of publically available data and subsequent data integration, poses useful for future metabolic modeling studies. However, the way in which differential expression was calculated for these 12 unique datasets, and the heavy reliance on in vitro data, when in vivo data is available, poses concerns in regards to the final selected metabolic targets.

Major points

- The use of 3 different algorithms for differential expression identification for the bulk RNA-seq datasets creates discontinuity in differential expression definitions. This then biases later GSEA, as well as differential expression comparisons across datasets.
- The use of Recon1 instead of a more updated model due to computational limitations is understandable, however steps should be taken to create a more accurate COVID-19 contextualized model, such as incorporating airway epithelial cell specific transport reactions, in addition to specific inflammatory pathways that are integral to COVID-19 disease mechanism.
- Why are both in vitro and in vivo data utilized? How are the confounding factors of gene expression in a dish vs. a human considered? Cell lines do allow for target validation, as seen from your remdesvir treated Vero E6 cells, however why is this in vitro data integrated during the initial DE, GSEA, metabolic modeling portion of the analysis pipeline? Especially when utilizing a whole-body based model such as Recon1, while integrating tissue specific data from multiple tissue types.

Minor points

- There are several more recent transcriptomic data integration methods...why weren't other such approaches explored for this analysis?
- The structure of Figure 4 is not consistent with other figures.

Reviewer #1:

Cheng and his colleagues performed genome-scale metabolic modeling observations for SARS-CoV-2 infected host cells to identify possible antiviral targets for COVID-19. Specifically, they built genome-scale metabolic models using 12 publicly available transcriptome data for SARS-CoV-2 infected host cells. These observations reveal new host metabolism reprogramming during SARS-CoV-2 infections and offered a few potential host factors related to SARS-CoV-2 infections, which were validated by published CRISPR-Cas9 screening data. Via analyzing RNA-sequencing data of remdesivir-treated Vero E6 cell samples, they prioritized several metabolic targets in possible combination with remdesivir. Although *in silico* findings were not validated by experimental assays, this study presented a new *in silico* methodology for development of COVID-19 therapeutic strategies from host metabolism perspectives. The manuscript is clearly written, the methods are described in sufficient detail and the conclusions are fairly supported by the data. Several specific comments need attention below, which may improve the manuscript further.

We appreciate the reviewer's positive comment on our study. We are pleased to note that we have now further performed siRNA assays to evaluate the antiviral efficacy of the knockdown of a set of predicted top targets, confirming many of them (following the request of Reviewer #2). These results are described in a new section in the main text titled "Validation of the predicted anti-SARS-CoV-2 targets with an *in vitro* siRNA assay" (Page 13), and a new Figure 4 describing these results. We appreciate the reviewer's other comments and have carefully addressed them below.

1. The authors inspected 12 publicly available transcriptome datasets for SARS-CoV-2 infected host samples. It will be interesting the authors can discuss cell type-specific and tissue-specific genome-scale metabolic models for SARS-CoV-2 infections, which will provide cell type-specific drug targets for effective therapeutic development.

We agree. Indeed, although the *in vitro* cell data among 12 datasets we analyzed in this study covers a few different tissue types of origin (e.g. lung, kidney, colon), this could be further studied in detail using the various single-cell datasets. We focused on the metabolism of epithelial cells from the single-cell datasets, but we agree that the metabolic changes of specific cell types, e.g. immune cells, during SARS-CoV-2 infection may be investigated in future studies. We have discussed this point as one of the limitations of our study (Discussion section, Page 21):

"This study has several limitations that should be more thoroughly addressed in future studies incorporating the GEM modeling approach. First, as we aimed to identify targets in airway epithelial cells, we did not fully characterize cell type and tissue-specific metabolism associated with SARS-CoV-2 infection. Notably, future studies should analyze single-cell datasets to construct cell type-specific GEMs to identify cell type-specific antiviral targets and virus-induced alterations, e.g, immunometabolic changes (O'Carroll et al. 2021)."

2. The authors identified several interacting metabolic pathways, such as Branched-chain amino acid (valine, leucine). Could the authors discuss possibility of SARS-CoV-2 infection mediated metabolism is sex- or age-specific. For example, branched-chain amino acid pathway is highly age-specific. It will help to better understand the age-specific morbidity in COVID-19 from genome-scale metabolic modelings.

We appreciate this interesting comment from the reviewer. Indeed, COVID-19 infection rate and morbidity are highly variable in different sex or age groups and in people with baseline metabolic diseases such as diabetes. It is possible that the different metabolic states associated with aging underlie some of the observed age-specific differences of COVID-19 epidemiology, which requires further studies. As to branched-chain amino acid metabolism, we have found that this particular finding is not robust to the technical variations in the metabolic modeling methods. Specifically, when we followed the suggestion of another reviewer to use a newer Recon 3D human metabolic model instead of the Recon 1 model, this metabolic pathway no longer appears as a highly consistent differential pathway (please refer to our reply to the Comment #2 of Reviewer #2 for details if of interest). Therefore, we have now removed the results on branched-chain amino acid metabolism. Nevertheless, we have now included a discussion of the possibility of sex or age-specific metabolic response to SARS-CoV-2 infection, together with the discussion of disease severity-specific metabolic changes. Please see our reply to the next comment for the updated text.

3. Children and older individuals have different responses to SARS-CoV-2 infectious. Individuals with SARS-CoV-2 infectious have different outcomes, including symptomatic vs. asymptomatic, mild, moderate and severe outcomes. It will be interesting the authors can discuss COVID-19 outcome-specific genome-scale metabolic models, which will help us understand disease severities of COVID-19 from metabolism perspectives.

As mentioned above, we agree that the age-specific response to SARS-CoV-2 infection could have metabolic underpinnings that needs to be studied further. We also agree that the degree and patterns of metabolic disruption are likely different in patients with different disease severity, which may be correlative or causal, and this also requires careful future investigation. Addressing this and the previous comment, the relevant Discussion section now reads as follows (Page 21):

“It’s known that people with different sexes and ages respond differently to SARS-CoV-2 infection (Peckham et al. 2020; Canas et al. 2021), and that the infection can result in a wide spectrum of disease severity (Sandoval et al. 2021), which may be associated with metabolic underpinnings that can in turn be studied in the future with sex, age and clinical outcome-specific GEMs, given sufficient pertaining preclinical and clinical data.”

4. Several related literatures may be discussed further, such as doi: 10.1371/journal.pbio.3000970 and doi: 10.1016/S2589-7500(20)30192-8.

Thanks. We have now cited these relevant studies in our manuscript (Page 21), as follows:

“Our GEM-based pipeline thus complements other computational methods in predicting anti-SARS-CoV-2 targets and drugs, including those based on network analysis (Zhou et al. 2020) or artificial intelligence (Zhou et al. 2020).”

5. Overall, this is a novel, computational study from SARS-Cov-2 host metabolism perspective. The authors may discuss future works, about metabolic synthetic lethality and metabolic vulnerability studies, for emerging development of host-targeting therapies for COVID-19.

We thank the reviewer for this insightful comment. Indeed a synthetic lethality-based approach has been proposed for antiviral infection (Mast et al. 2020). We included a brief discussion of this point in the Discussion section (Page 22):

“While we predicted combinations of targets based on the principle of restoring cellular homeostasis, it is also feasible to predict synergistic target combinations under the GEM framework via modeling of synthetic lethality, which has also been proposed as a viable antiviral approach (Mast et al. 2020).”

Reviewer #2:

The authors presented an integrated analysis of 12 published in vitro and human patient gene expression datasets on SARS-CoV-2 infection using genome-scale metabolic modeling (GEM) and revealed the reprogramming of host metabolism during SARS-CoV-2 infection. They also applied the GEM based metabolic transformation algorithm to predict anti-SARS-CoV-2 targets that counteract the virus-induced metabolic changes and validated them using published experimental drug and genetic screen data. Moreover, they generate and analyzed RNA-sequencing data of remdesivir treated Vero E6 cell samples and predicted metabolic targets acting in combination with remdesivir. They provided top high-confidence candidate targets with support from human patient data for evaluation in further studies. This is a very interesting study; however, it lacks experimental evidence for the targets identified in the study. I strongly suggest the authors validate their top predictions by performing additional experiments.

We thank the reviewer for the interest in our study. We agree that experimental validation of the predicted targets can greatly boost the value of our study. Accordingly, we have further performed siRNA assays to evaluate the antiviral efficacy of the knockdown of a set of predicted top targets, which showed encouraging positive results (please see our reply to the Comment #3 below for details). We have also carefully addressed your other comments, as described below.

Major Comments

1. Flux balance analysis renders a considerable part of the result and contributes significantly to the main conclusion and discovery in this study. However, there is some missing information about how the modeling is performed and what rationality is. First of all, there is no information about constraints used in the modeling, which is a very important part of flux balance analysis and could significantly affect the final results. It could be that the author used the default

constraints from Recon1, but it then raises another very serious concern if the rationale is to use the same constraints to model *in vitro* cell lines and *in vivo* human cells. At the very beginning, it may be useful to compare the fluxes between cell lines and human tissues as the rate of the flux is expected to be at a different scale.

We apologize for the confusion. We realized that our description of metabolic modeling methods was not entirely clear and have updated the relevant writing (see below). Actually, flux balance analysis (FBA) was not used in our study. Our metabolic modeling was based on the iMAT algorithm (Shlomi et al. 2008), which used mixed integer programming to find the optimal adjustment of reaction bounds such that reactions with high fluxes and low fluxes correspond to high and low gene expression, respectively, in a sample type and dataset-specific manner. Therefore, the optimal and yet different constraints for different *in vitro/in vivo* datasets were determined by iMAT to best fit the context-specific gene expression data. We did NOT apply FBA to the resulting constrained model, as maximal biomass production may not be appropriate especially for the *in vivo* patient nasopharyngeal swab and BALF samples, and tissue type-specific objective functions for these samples are very difficult to define rigorously. Instead, the expression-based constrained model generated by iMAT was randomly sampled to obtain a distribution of flux values for each reaction. We then were able to determine reactions with differential fluxes by comparing the flux distributions of a reaction in control and virus-infected conditions following the procedure described in (Shlomi et al. 2008). Besides, we did not attempt to normalize the raw expression values across datasets (this is very difficult due to large platform differences and batch effects confounded by biological factors; therefore this approach does not allow the comparison of the **base-level** expressions across datasets). Instead, we aimed to identify the SARS-CoV-2-induced **differential** expressions and **differential** fluxes within each dataset, and then compared these **differential** results qualitatively across datasets, as our key question is about the SARS-CoV-2-induced changes rather than base-level metabolism.

We have now revised the Results section and Materials and Methods section accordingly, aiming to clarify our approach:

Results section, under the sub-section "Genome-scale metabolic modeling (GEM) identifies SARS-CoV-2-induced patterns of metabolic flux changes", Page 7:

"Briefly, iMAT uses mixed integer programming to optimally identify high and low-activity reactions that match the high and low gene expression patterns in a sample-specific manner, thus defining sample-specific model constraints to obtain contextualized models (Shlomi et al. 2008). ... After obtaining the dataset and sample-specific constrained model with iMAT, the marginal distribution of flux values of each metabolic reaction was obtained by sampling. The flux distributions of the control and infected groups were compared and reactions with differential fluxes (DF) were identified (Methods; Table S4)."

Materials and Methods section, under the sub-section "Computation of metabolic fluxes from gene expression data with genome-scale metabolic modeling", Page 23:

“Although not a single value representing a unique solution, we were able to determine reactions with differential fluxes by comparing the flux distributions of a reaction in control and virus-infected conditions (see next section). We did not apply flux balance analysis (FBA) on the iMAT-derived constrained models, as maximal biomass production may not be appropriate especially for the in vivo patient samples, and tissue type-specific objective functions for these samples are not trivial to define.

2. Recon1 is used for the FBA in this study, and the author claimed this is because using more recent models will significantly increase the computational cost. This is not acceptable since there are many different updated models since 2017. Recon 3 and Human1 can be used. This is also an unfair statement, as the scale between Recon1 and the most recent GEM Recon 3 and Human1 is not significant and the computational cost for running iMAT using these models will not be that different. In addition, Recon1 has many mass imbalanced reactions which could affect the FBA results and lead to incorrect conclusions. So, I would argue that the authors should use one of the most recent human GEM for the FBA analysis.

We thank the reviewer for this comment and agree that it's better to test our analysis and findings using a more up-to-date genome-scale metabolic model, which also allows us to evaluate the performance of the MTA/rMTA algorithm on a larger model (they mostly have not been tested on the more recent large-scale metabolic models with >10k reactions). We therefore redid all the metabolic modeling analysis in our study with the Recon 3D model (Brunk et al. 2018) and have updated our reported results throughout (results corresponding to the previous Figure 2, 3 and 4 have all been updated). Overall, we obtained similar results on SARS-CoV-2-induced metabolic changes as well as on antiviral target prediction using Recon 3D compared to Recon 1, and our major conclusion remains unchanged. However, some of the results differ in the details, and a few results from some of the minor analyses were not reproduced with Recon 3D; we hence removed the non-robust results from our manuscript. We next provide more details below.

First, encouragingly, we confirmed that with Recon 3D, the rMTA predictions still got well validated using the published resources of known anti-SARS-CoV-2 targets (although naturally the numerical results differ from our previous ones obtained with Recon 1). These validation results include: 1. Predictions from 9 out of the 12 datasets had significant enrichment (FDR<0.05) with the targets identified in CRISPR-Cas9 genetic screens (Wei et al. 2021 and Daniloski et al. 2021; Figure 3B); 2. Based on the CRISPR-Cas9 genetic screen data, rMTA predictions in 6 out of the 12 datasets achieved AUROC>0.6 and as high as 0.72 (Figure 3E); 3. The predictions from some datasets were enriched for various reported experimentally validated drug targets (compiled by Kuleshov et al. 2020; Figure 3C); 4. The predictions from some datasets are also enriched for host proteins identified to interact with SARS-CoV-2 proteins (from Gordon et al. 2020 and Stukalov et al. 2020; Figure 3D). Further, many of the metabolic pathways enriched by the updated predictions were also similar to those obtained previously with Recon 1, for example, extracellular transport reactions and inositol phosphate metabolism were both enriched (Figure 3G). However, the results on the specific enrichment of extracellular transport reactions by each metabolite species with Recon 3D showed a large difference compared to Recon 1, and these enrichment results were hence removed. Other numerical

differences in the results exist, as one would expect. E.g. the rMTA prediction performance on some of the datasets changed -- the prediction based on the BALF dataset (Xiong et al. 2020) exhibited a better AUROC now with Recon 3D (0.62) vs with Recon 1 (0.49), although the best AUROC we obtained across all datasets decreased (0.81 with Recon1 to current 0.72 with Recon 3D; Figure 3E,F). These suggest that while predictions on individual dataset can be somewhat sensitive to the metabolic model being used, the rMTA algorithm overall exhibited a capacity to identify potentially biologically meaningful targets. We attach the updated **Figure 3** below for your convenience, and have updated the corresponding Results section (mostly modifying numerical results such as P values) under "*Prediction of anti-SARS-CoV-2 targets that act via counteracting the virus-induced metabolic changes*".

Figure 3. Genome-scale metabolic modeling (GEM)-based prediction of anti-SARS-CoV-2 targets that act via reversing the virus-induced metabolic alterations. The robust metabolic transformation algorithm (rMTA, Valcárcel et al. 2019) was used to predict metabolic reactions whose knock-out can reverse the SARS-CoV-2-induced metabolic changes using each of the collected datasets (Methods). (A) Visualization of the overlap of the top 10% MTA-predicted target reactions between each pair of datasets analyzed using Fisher's exact tests (Methods). The dot size corresponds to the

effect size of the overlap as measured by odds ratio, and the color corresponds to the negative log₁₀ adjusted one-sided P value (grey means below 0.05). **(B)** A summary visualization of the enrichment of the top 10% MTA-predicted targets from each dataset for the antiviral hits (i.e. those whose KO inhibits SARS-CoV-2 infection) identified in the two published CRISPR-Cas9 screens (Wei et al. 2021 and Daniloski et al. 2021; Methods). Y-axis represents the odds ratio of enrichment, the dot color corresponds to the negative log₁₀ adjusted one-sided P value from Fisher's exact tests, and dot size corresponds to the number of enriched target reactions. The datasets are ordered by P values, the first 8 datasets have FDR<0.1. **(C)** cases of significant enrichment (FDR<0.1) of top 10% MTA-predicted targets from each dataset for the experimentally validated anti-SARS-CoV-2 drug sets from previous studies (compiled by Kuleshov et al. 2020). "Union" means the union of all drug sets. Axes and the meanings of dot color and size are similar to (B) but the axes are flipped and the adjusted P values are not log-transformed. **(D)** significant cases of significant enrichment (FDR<0.1) of top 10% MTA-predicted targets from each dataset for the host proteins involved in host-SARS-CoV-2 protein-protein interactions (combined from Stukalov et al. 2020 and Gordon et al. 2020). Axes and the meanings of dot color and size are similar to (C). **(E)** A summary of the area under ROC curve (AUROC) value of the MTA prediction using each dataset, based on positive and negative sets (i.e. genes whose KO inhibits or promotes the viral infection, respectively) from the two published CRISPR-Cas9 screen data (Wei et al. 2021 and Daniloski et al. 2021; Methods). The error bars (vertical lines through the dots) represent 95% confidence intervals. **(F)** Example ROC curves from two of the best-performing datasets, one in vitro (Vero) and one in vivo (BALF). The curves are colored by color gradients corresponding to the threshold of top MTA predictions. **(G)** Summary of the metabolic pathways significantly enriched (FDR<0.1) by the final list of consensus candidate targets identified across datasets (Methods). The axes and the meanings of dot color and size are similar to (B) except for that the axes are flipped.

Second, the metabolic pathways found to be enriched with metabolic reactions with consistently differential fluxes following SARS-CoV-2 infection are also similar to those originally reported with Recon 1, including various metabolite transport reactions, pentose phosphate metabolism, pyrimidine synthesis, fatty acid synthesis, etc. (Figure 2B), although the TCA cycle no longer reaches statistical significance and is now removed. For clarity of visualization, we simplified the pathway diagrams based on the Recon 1 network as the network of Recon 3D contains many more edges. The highlighted flux alteration patterns also remain largely the same for pyrimidine synthesis, inositol phosphate metabolism, and fatty acid synthesis pathways (previous Figure 2D,F,H). In contrast, we did not reproduce the Recon1 findings on the differential fluxes in the metabolite transport across the cell membrane (previous Figure 2C) and hence removed this analysis. The updated Figure 2 is attached below for your convenience, and we have also updated the corresponding Results section (mostly numerical results such as P values) under "Genome-scale metabolic modeling (GEM) identifies SARS-CoV-2-induced patterns of metabolic flux changes".

Figure 2. Genome-scale metabolic modeling (GEM)-based analysis of SARS-CoV-2-induced metabolic alterations across datasets. GEM was used to compute the metabolic fluxes from the gene expression profiles, and reactions with differential fluxes (DF) between the SARS-CoV-2-infected and control groups were identified for each dataset (Methods). **(A)** Visualization of the overlap of the top DF reactions between each pair of datasets analyzed using Fisher's exact tests (Methods). The dot size corresponds to the effect size of the overlap as measured by odds ratio, and the color corresponds to the negative log10 adjusted one-sided P value (grey means below 0.05). **(B)** A summary visualization of the metabolic pathway enrichment result for the top consistent DF reactions across the datasets, with more importance given to the various in vivo patient datasets (Methods). Y-axis represents the odds ratio of enrichment, the dot color corresponds to the adjusted P value from Fisher's exact tests, and dot size corresponds to the number of enriched reactions within each pathway. Half-dots plotted on the top border line correspond to infinity odds ratio values. The pathways on the X-axis are ordered by P value and only those with $FDR < 0.1$ are shown. **(C-E)** Visualization of the relatively consistent DF patterns in selected enriched pathways. The DF results are based on metabolic modeling using the human GEM Recon 3D (Brunk et al. 2018), but for clear visualization, the metabolic network graphs are based on the human GEM Recon 1 (Duarte et al. 2007) to reduce the number of metabolites and reactions displayed (Methods). Metabolites are represented by nodes, reactions are represented

by directed (hyper) edges, with edge direction corresponding to the consensus reaction direction and edge color corresponding to the consensus DF direction across datasets (Methods). Red and blue colors correspond to increased and decreased fluxes, respectively; grey color corresponds to reactions not showing consistent DF changes across datasets, some of such reactions are not shown to increase clarity. **(C)** Pyrimidine synthesis. **(D)** Inositol phosphate metabolism. **(G)** Glycine, serine, and threonine metabolism. **(E)** Fatty acid synthesis. Metabolites are labeled by their names in (C) or IDs in (D,E), with suffixes denoting their cellular compartments: [c] cytosol; [m] mitochondria. The mapping between the IDs and metabolite names in (D,E) is given in Table S5D.

Third, we also updated the prediction of combinatory targets with remdesivir using Recon 3D (previous Figure 4, now reported in Figure 5). Again, the results are overall similar to those obtained from Recon 1 with only minor differences. As reported with Recon1, remdesivir did not fully reverse the metabolic changes induced by SARS-CoV-2 (Figure 5C), the pathways enriched by differential flux reactions comparing the virus+remdesivir group vs the control group (Figure 5D) and the pathways enriched by the rMTA-predicted targets for combination with remdesivir (Figure 5E) remained similar. Figure 5 is attached below for your convenience, and have updated the corresponding Results section (mostly numerical results such as P values) under "Prediction of metabolic targets for anti-SARS-CoV-2 in combination with remdesivir".

Figure 5. Analysis of the gene expression and metabolic flux profile of remdesivir treatment and prediction of metabolic targets for anti-SARS-CoV-2 in combination with remdesivir. (A) PCA plot of the gene expression profiles for the Vero E6 samples from all experimental groups: control (no virus or remdesivir treatment), virus (SARS-CoV-2-infected), virus+remdesivir (SARS-CoV-2-infected treated by remdesivir), remdesivir (remdesivir treatment alone without virus). There are 3 replicates in each group. (B) A visualization of selected differentially expressed pathways comparing the virus+remdesivir group to the control group using gene set enrichment analysis (GSEA).

Y-axis represents normalized enrichment score (NES), positive value means higher expression in the virus+remdesivir group compared to control, vice versa. Pathways on the X-axis are ordered by their NES values, all pathways displayed have adjusted $P < 0.05$; pathways with names in red are those that are not significantly different (adjusted $P > 0.2$) when comparing the virus group to the control group using GSEA, i.e. changes in these pathways may arise from remdesivir-specific effects. The dot color corresponds to the negative \log_{10} adjusted GSEA P value, and dot size corresponds to the number of enriched genes (i.e. “leading edge” genes in GSEA). (C) PCA plot of the average metabolic flux profile computed using the iMAT algorithm (Shlomi et al. 2008; Methods) representative of each of the experimental groups, the labels are the same as (A). (D) A visualization of metabolic pathway enrichment results of the differential metabolic fluxes in the virus+remdesivir group vs the control group, using Fisher’s exact tests (Methods). Y-axis represents the odds ratio of enrichment, the horizontal dashed line corresponds to odds ratio of 1. The dot color corresponds to the negative \log_{10} adjusted one-sided P value from Fisher’s exact tests, and dot size corresponds to the number of enriched target reactions. The datasets are ordered by P values, all pathways displayed have $FDR < 0.1$. (E) The robust metabolic transformation algorithm (rMTA, Valcárcel et al. 2019) was used to predict metabolic reactions whose knock-out can further transform the metabolic state of the remdesivir-treated SARS-CoV-2-infected cells back to the normal control state, using the Vero E6 cell samples (Methods). The significant metabolic pathways ($FDR < 0.1$) enriched by the top 10% MTA-predicted targets are shown. Axes and the meanings of dot color and size are similar to (D).

We would like to note that in comparison to Recon 1, the new modeling runs with Recon 3D take about 100x more CPU-hours, even though Recon 3D only contains about 3x more reactions and 2x more metabolites than Recon 1. This arises since the iMAT and rMTA algorithms used in our study solve a large number of mixed integer programming (MIP) and quadratic programming (QP) problems, which are extremely computation extensive with non-linear time complexity. Given this, we recommend using a high-performance computing cluster with parallelization for readers interested in applying our pipeline. The MIP and QP may also have numerical instability issues with large models like Recon 3D. Thus, as described above, we did not rely solely on the results of either Recon 3D or Recon 1 but combined the results of both to identify the most robust findings.

3. One of the very important contributions of this study is that the author proposed some promising targets that could be used together with remdesivir for COVID19 treatment based on their in-house in vitro experimental data. This means their lab is capable of performing COVID19 in vitro tests. However, I am surprised that they have not tested the effect of the targets (top candidate(s)) they identified using siRNA or plasmid as that should be relatively simple compared to what they have already done. The conclusion of this study will be significantly enhanced if the authors perform this experiment. The readers of the MSB would also expect to see the validation experiments.

We thank the reviewer and agree that experimental validation of the predicted targets can greatly boost the value of our study. We have therefore further performed siRNA assays to evaluate the antiviral efficacy of the knockdown of a set of predicted top targets. Briefly, we transfected Caco-2 cells with siRNAs targeting each of our top predicted targets, we infected these cells with SARS-CoV-2, and using immunofluorescence staining for SARS-CoV-2 N protein we evaluated the antiviral efficacy of the predicted targets. Knocking down the top targets showed significant reduction in SARS-CoV-2 replication compared to negative controls (non-targeting scrambled siRNAs), while not showing any significant cytotoxicity (i.e. no reduction in cell number as determined by DAPI staining). We have added a new section titled "Validation of the predicted anti-SARS-CoV-2 targets with an in vitro siRNA assay" (Page 14), and a new Figure 4 describing these results. These are also attached below for your convenience.

"Validation of the predicted anti-SARS-CoV-2 targets with an in vitro siRNA assay"

We next seek to experimentally validate the consensus predictions using an *in vitro* siRNA-based target knock-down assay. We have previously conducted a genome-wide siRNA screen to identify host factors that are essential for SARS-CoV-2 replication (Methods). To further prioritize targets for validation, we applied our computational predictions to this dataset and selected a small subset of 39 genes among the 81 consensus predicted targets (Methods; target list given in Table EV7C). siRNAs targeting each of these 39 genes were individually transfected into Caco-2 cells (n=4), which were then infected with SARS-CoV-2. Viral replication at 48 hours post-infection was assayed with immunofluorescence labeling of the viral nucleoprotein (N) protein, and siRNA-mediated toxicity was evaluated with DAPI staining (illustrated in **Figure 4A**; Methods). Overall, compared to negative control non-targeting siRNAs (scrambled), we observed that siRNAs targeting the 39 consensus targets significantly reduced viral replication (Wilcoxon rank-sum test $P=1.7e-9$, **Figure 4B**; raw data in Table EV7D; Methods). Inspected individually, knocking down 34 of the 39 targets significantly reduced SARS-CoV-2 replication (adjusted $P<0.05$, Table EV7E). Notably, we also evaluated 4 randomly selected metabolic genes that were not predicted to revert SARS-CoV-2-induced metabolic changes using our analyses (negative controls), which showed much weaker viral inhibition effects (blue dots in **Figure 4B**). *ACE2* and *TMPRSS2*, two genes known to be essential for SARS-CoV-2 cellular entry (Hoffmann et al. 2020), were included as positive controls for comparison (red dots in **Figure 4B**). Overall, knock-down of the predicted consensus targets did not significantly reduce cell number ($P=0.73$, **Figure 4C**), indicating that their impact on viral replication is likely not due to siRNA-mediated cytotoxic effects. Representative fluorescence microscopy images for the siRNA targeting the top three predicted targets (together with the scrambled non-targeting control and *ACE2* as positive control) are shown in **Figure 4D**. These results experimentally validate the efficacy of our predicted targets *in vitro*.

Figure 4. Validation of the predicted anti-SARS-CoV-2 targets with an immunofluorescence-based *in vitro* siRNA assay. (A) A schematic illustration of the siRNA assay in Caco-2 cells infected with SARS-CoV-2 to validate the antiviral efficacies of the consensus predicted metabolic targets. Caco-2 cells were transfected with siRNAs

for 48 h prior to infection with SARS-CoV-2 (MOI=0.1). Four replicates were performed for each target. At 48 h post-infection, cells were subjected to staining with SARS-CoV-2 nucleoprotein (N) antibody and DAPI, and then imaged to determine the percentage of infected cells after each target knockdown (Methods). **(B)** Quantification of SARS-CoV-2 infection. After the siRNA knockdown of each target, viral infection was quantified as mean \log_2 fold-change (\log_2 FC) of the percentage of SARS-CoV-2⁺ cells relative to the mean of scrambled non-targeting siRNAs (“SCRAMBLED”, green dots; accordingly the mean \log_2 FC value of scrambled non-targeting siRNAs was normalized to zero). Predicted positive targets (“TARGETING”, grey dots), predicted negative targets (“NEG”, blue dots) and positive controls (“POS”, red dots, including *ACE2* and *TMPRSS2*) are all shown. Wilcoxon rank-sum test P value comparing the predicted positive targets to scrambled non-targeting siRNAs is given. **(C)** Quantification of cell number after each target knockdown, calculated as the mean fraction of DAPI⁺ cells relative to the scrambled non-targeting siRNAs (i.e. the latter was normalized to 1). Colors of dots and P value are interpreted in the same way as in (B). **(D)** Representative fluorescence images from the siRNA assay showing SARS-CoV-2 infection (green channel, top row), cell number (blue channel, middle row) and merged (bottom row). Results for scrambled non-targeting siRNA as negative control (left column), knockdown of three predicted top metabolic targets (PIKFYVE, SLC16A10, and PIP5K1C, middle columns), and knockdown of positive control (*ACE2*, right column) are shown. Scale bar=10 μ m.

4. There are many parameters proposed by the authors arbitrarily selected and need sensitivity analysis. For example, the authors selected 'top 400' DE genes with at least 'FDR <0.1' for all different COVID19 datasets, which seems a very sensitive cutoff. Why 400? Will the result change significantly if you choose top 500? Also, the authors selected 'FDR<0.2' for metabolic pathway meta-analysis, which is neither statistically sound nor consistent with the other cutoff used in this study.

We agree that in these two particular cases, the cutoffs used were arbitrary or not stringent enough. We have now switched to using FDR<0.1 for both cases, which is consistent with all the other analysis in our study and we believe is a more generally acceptable threshold. More details are provided below.

In the case of DE genes, the previous cutoff of n=400 was *only* used in Figure 1B to measure the extent of overlap between each pair of datasets. We previously decided to use a constant number so that the odds ratio calculation had a more uniform denominator, and 400 is just below the number of FDR<0.1 DE genes in the dataset with the fewest DE genes (whereas other datasets can have thousands of DE genes at FDR<0.1). Now, following your comment, we have updated Figure 1B with the fixed FDR<0.1 cutoff (attached below). The range of odds ratio and P values changed as expected, but the clustering of datasets largely remains the same, e.g. the various single-cell datasets clustered together; 293T, A549 and Vero clustered together; and Calu-3, NHBE and the Swab.Lieberman datasets clustered together. Further robustness analysis with additional DE cutoffs also produced similar results (now provided in

Appendix Figure S1). The description of numerical results (such as P values) in the section “Integrated analysis of multiple gene expression datasets identifies coherent immune and metabolic changes in SARS-CoV-2 infection” (Page 4) has been updated accordingly.

Updated Figure 1B:

Figure 1. (B) Visualization of the overlap of the top significant DE genes (FDR<0.1) between each pair of datasets analyzed using Fisher's exact tests (Methods). The dot size corresponds to the effect size of the overlap as measured by odds ratio, and the color corresponds to the negative log10 adjusted one-sided P value (grey means below 0.05).

We also changed the cutoff to FDR<0.1 in the case of the metabolic pathway meta-analysis, and the corresponding Figure 1E has been updated (also attached below; the previous version is also attached). The overall pattern of metabolic changes remains similar regardless of the cutoff used.

Updated Figure 1E:

Figure 1. (E) Heatmap summarizing the landscape of metabolic pathway alterations (based on gene expression) during SARS-CoV-2 across datasets. The heatmap color corresponds to the GSEA NES values (explained above) for KEGG metabolic pathways grouped into major categories. Only the metabolic pathways with FDR<0.1 enrichment in at least one dataset are included in the heatmap. The dataset labels used in this figure correspond to those given in Table 1.

5. The author claimed that they used an in-house R package named 'gembox' for the GEM based analysis. The authors should provide the script for the analysis to enable proper assessment.

Thanks. We have now made this package available to the general public on GitHub. We have added the relevant information to Data and Code Availability statement, Page 29:

Our in-house R package named gembox used for all the GEM analysis in this study can be found on GitHub: <https://github.com/ruppinlab/gembox>.

Reviewer #3:

Summary

Cheng et al utilizes 12 published SARS-CoV2 gene expression data sets to investigate virus-induced host metabolic changes, using genome scale metabolic modeling. Based on these identified metabolic shifts, they then identified potential metabolic drug targets using MTA, and compared these proposed metabolic targets to published drug and genetic screening data.

Lastly, they investigated potential coupled therapeutics through their remdesivir-treated Vero E6 cell RNA-seq data. The goal of this paper is to utilize genome scale metabolic modeling to identify potential host treatment avenues for SARS-CoV2 infection, as opposed to the more common approach of identifying viral targets. This unique approach to viral-infection therapeutic development creates a pipeline for drug target discovery that could be applied to other viral infections outside of SARS-CoV2. Additionally the use of publically available data and subsequent data integration, poses useful for future metabolic modeling studies. However, the way in which differential expression was calculated for these 12 unique datasets, and the heavy reliance on *in vitro* data, when *in vivo* data is available, poses concerns in regards to the final selected metabolic targets.

We appreciate the reviewer's positive comments on the value of our study. We apologize if it was not sufficiently explicated previously, but indeed, we have collected and analyzed 12 datasets of SARS-CoV-2 infection, 6 of which are *in vivo* datasets of samples from human patients. The other 6 *in vitro* datasets include samples from primary human bronchial epithelial cells as well as immortalized or cancer cell lines. We think this is a good balance, as including *in vitro* datasets could help bridge pre-clinical and future clinical drug development -- for example, following the request of Reviewer #2, we have now performed an siRNA assay *in vitro* in Caco-2 cells and successfully validated our consensus list of predicted targets (see reply to Reviewer #2 Comment #3 for details), which we hope can form a solid foundation for further investigation and downstream drug development. Having said that, we should indeed emphasize that our final list of selected targets focuses on the top predictions that are also supported by the analysis of the *in vivo* datasets, which showed good validation results. To summarize, we identify *in vitro* arising candidates that are supported also by the *in vivo* data analysis and hence more likely to work in patients. Please kindly refer to our reply to the Comment #3 for more details.

Major points

1. The use of 3 different algorithms for differential expression identification for the bulk RNA-seq datasets creates discontinuity in differential expression definitions. This then biases later GSEA, as well as differential expression comparisons across datasets.

We thank the reviewer for this comment. Indeed, different differential expression (DE) algorithms normalize the expression data in very different ways and use different statistical models. Naturally, these differences will lead to numerically different results. We therefore focused on verifying the robustness of our major conclusion to the choice of DE method. The major conclusion from our DE analysis is that a large number of metabolic pathways are consistently altered by SARS-CoV-2 infection, representing a major domain of cellular molecular changes besides immune response (i.e. Figure 1D). We repeated the DE analysis using the limma-voom method (Law et al. 2014) for *all* the bulk RNA-seq datasets, then repeated the pathway enrichment analysis. We found that despite the expected numerical differences, the consistently enriched DE pathways across datasets did not change much -- like before, we still see that the most consistent positively enriched (upregulated) pathways include various immune response-related pathways, and the most consistent negatively enriched (downregulated)

pathways represent many different metabolic pathways, and the particular individual enriched pathways were also highly similar. Therefore the previously reported pertaining findings remain robust to the choice of DE methods. The new result with limma-voom is now included as Appendix Figure S2, which is also attached below for your convenience. For comparison, we also attach the Figure 1D below. We have added a sentence to the corresponding Results section “Integrated analysis of multiple gene expression datasets identifies coherent immune and metabolic changes in SARS-CoV-2 infection” (Page 5) as follows: “The major findings above are robust to the DE algorithms used (Appendix Figure S2).”

Appendix Figure 2, results based on limma-voom for all bulk RNA-seq datasets:

Appendix Figure S2. Pathway enrichment analysis of SARS-CoV-2-induced differential changes with the alternative differential expression (DE) algorithm limma-voom. A summary visualization of the GSEA result for the top consistently altered pathways during SARS-CoV-2 infection across the datasets, with more importance given to the various *in vivo* patient datasets (Methods). Unlike Figure 1D in the main text where the bulk RNA-seq dataset DE results were obtained using mostly DESeq2 mixed with other algorithms, here limma-voom was used for the DE analysis of all bulk RNA-seq datasets. The dot color corresponds to the negative log10 adjusted P values from GSEA, with two sets of colors (red-orange and blue-purple) distinguishing up-regulation from down-regulation (positive or negative normalized enrichment scores, i.e. NES); dot size corresponds to the absolute value of NES measuring the strength of enrichment. The left and right-hand side blocks represent the pathways that tend to be consistently up-regulated and down-regulated in infected vs control samples, respectively; within each block, the pathways are ordered by negative sum of log P values across datasets (i.e. Fisher’s method).

Figure 1D, results based on mostly DESeq2 for bulk RNA-seq datasets:

Figure 1. (D) A summary visualization of the GSEA result for the top consistently altered pathways during SARS-CoV-2 infection across the datasets, with more importance given to the various *in vivo* patient datasets (Methods). The dot color corresponds to the negative log10 adjusted P values from GSEA, with two sets of colors (red-orange and blue-purple) distinguishing up-regulation from down-regulation (positive or negative normalized enrichment scores, i.e. NES); dot size corresponds to the absolute value of NES measuring the strength of enrichment. The left and right-hand side blocks represent the pathways that tend to be consistently up-regulated and down-regulated in infected vs control samples, respectively; within each block, the pathways are ordered by negative sum of log P values across datasets (i.e. Fisher’s method).

We note that between DESeq2 (our previous DE algorithm used for most of the datasets) and limma-voom though, we prefer the former as it more directly models the distribution of the discrete read counts rather than utilizing a transformation to force the expression values to conform to distributions proper for a Gaussian-based model, and has shown good performance in some benchmark studies (e.g. Costa-Silva et al. 2017). Therefore in our main text, we still retained the use of DESeq2 as much as possible. The description of numerical results in the section “Integrated analysis of multiple gene expression datasets identifies coherent immune and metabolic changes in SARS-CoV-2 infection” (Page 5) have been very minorly updated due to DE method change to DESeq2 for the Weingarten-Gabby et al. (293T) dataset.

Reference:

Costa-Silva J, Domingues D, Lopes FM. RNA-Seq differential expression analysis: An extended review and a software tool. PLoS One. 2017;12(12):e0190152.

2. The use of Recon1 instead of a more updated model due to computational limitations is understandable, however steps should be taken to create a more accurate COVID-19 contextualized model, such as incorporating airway epithelial cell specific transport reactions, in addition to specific inflammatory pathways that are integral to COVID-19 disease mechanism.

We would like to clarify that Recon 1 was used only as the basis for further model building in our study, which (exactly following the spirit of your suggestion) was further refined to obtain contextualized models in a sample type or dataset-specific manner based on the sample-specific gene expression profiles using the iMAT algorithm. iMAT uses mixed integer programming to find the optimal adjustment of reaction bounds such that reactions with high fluxes and low fluxes correspond to high and low gene expression, respectively, in a sample type and dataset-specific manner. In this way, SARS-CoV-2-infection contextualized metabolic flux descriptions were obtained. Recon 1 is a general human model that includes the union of metabolic reactions and metabolites existing in different types of human cells, which serves as a good basis for model contextualization with iMAT. As for inflammatory pathways, they are not considered metabolic pathways and do not conform to the metabolic steady state assumption, therefore they are not within the domain of genome-scale metabolic modeling and are not included in any of the available genome-scale metabolic models. To further clarify without repeating the description of iMAT (a very well known and highly cited algorithm in the GEM field), we have added the sentence below in the Results section, under the sub-section “Genome-scale metabolic modeling (GEM) identifies SARS-CoV-2-induced patterns of metabolic flux changes”, Page 7:

Briefly, iMAT uses mixed integer programming to optimally identify high and low-activity reactions that match the high and low gene expression patterns in a sample-specific manner, thus defining sample-specific model constraints to obtain contextualized models (Shlomi et al. 2008).

3. Why are both in vitro and in vivo data utilized? How are the confounding factors of gene expression in a dish vs. a human considered? Cell lines do allow for target validation, as seen from your remdesvir treated Vero E6 cells, however why is this in vitro data integrated during the initial DE, GSEA, metabolic modeling portion of the analysis pipeline? Especially when utilizing a whole-body based model such as Recon1, while integrating tissue specific data from multiple tissue types.

As mentioned above, the datasets we analyzed represent a balanced collection of 6 *in vivo* and 6 *in vitro* datasets. Each of the datasets were first analyzed individually and independently from each other, the results (DE, GSEA, and metabolic modeling) were then compared across the datasets, and further we aimed to identify and robust findings that are consistent across both *in vivo* and *in vitro* datasets. Specifically for the metabolic modeling, as explained in our reply to the previous comment, for each dataset, we actually obtained a contextualized model specific to that dataset, so that the metabolic information across multiple tissue types were not mixed together in a single modeling – as the referee has acutely observed that indeed would not make sense (and was not what we did). Indeed, the *in vivo* datasets are

in principle more clinically relevant, but we think that the *in vitro* datasets also have their importance and value. We include the *in vitro* datasets for the following reasons:

1. They facilitate the comparison between *in vitro* model systems and *in vivo* patient samples, and provide hints on the “goodness-of-fit” of the *in vitro* models to the biology of the viral infection in human patients.
2. By selecting the consensus targets that were predicted based on both *in vitro* and *in vivo* datasets, we aim to increase the robustness of the predictions. This strategy also facilitates the use of *in vitro* assays for testing and validating the predicted targets, hence bridging pre-clinical and future clinical drug development -- for example, following the request of Reviewer #2, we have now performed an siRNA assay *in vitro* in Caco-2 cells and successfully validated our consensus list of predicted targets (see reply to Reviewer #2 Comment #3 for details), which we hope can form a solid foundation for further investigation and downstream drug development.
3. As can be seen, there is quite a great abundance of *in vitro* datasets on SARS-CoV-2 infection. We believe that it would be beneficial to take advantage of the information present in these datasets to glean more biological insights. It would not be very reasonable and a waste of resources to completely ignore all the available *in vitro* datasets.

We have added these points above to the Discussion section (Page 19):

We tried to cover datasets on both popular *in vitro* models of SARS-CoV-2 infection as well as human patients (nasopharyngeal swab and BALF samples), aiming to increase the robustness and clinical relevance of our findings. This strategy may also facilitate the testing of our predicted targets and bridge pre-clinical and potential future clinical drug development

Minor points

- There are several more recent transcriptomic data integration methods...why weren't other such approaches explored for this analysis?

Indeed, there are a few different methods for meta-analysis of gene expression microarray or bulk RNA-seq datasets (e.g. Rau et al. 2014, Waldron et al. 2016, reviewed in Toro-Domínguez et al. 2021). In recent years, there has also been much progress in the integration of multiple single-cell RNA-seq datasets (reviewed in Argelaguet et al. 2021). However, the datasets we analyzed in this study represent a very complex collection of bulk RNA-seq, single-cell RNA-seq, as well as proteomics datasets. The so-called “multi-omic integration” is still a maturing field with many existing challenges, although many methods have been developed (e.g. surveyed in Pierre-Jean et al. 2020), we find a lack of methods and software specifically designed to integrate our collected types of datasets that are well-benchmarked for our particular aim of analysis. The existing methods are also not compatible with our downstream metabolic modeling analysis. We therefore resorted to the simpler yet

reasonable approach of summarizing and synthesizing results across datasets based on the P values, which is more straightforward and grounded in classical meta-analysis ideas. We have extended the discussion on this point in the Discussion section (Page 20), which is also attached below for your convenience:

In terms of methodology, our complex collection of data from a wide range of platforms with large technical variations (bulk RNA-seq, scRNA-seq, MS-based proteomics) poses a challenge to a formal effect size-based meta-analysis. Despite the progress in multi-omic data integration (Pierre-Jean et al. 2020), to the best of our knowledge there is currently no method specifically for integrating the data types we used in this study that are also compatible with our downstream metabolic modeling. Therefore, we instead relied mostly on P values, and made subjective decisions that give higher importance to the various patient datasets when defining consistent findings, aiming to obtain results of higher clinical relevance.

References:

Rau A, Marot G, Jaffrézic F. Differential meta-analysis of RNA-seq data from multiple studies. *BMC Bioinformatics*. 2014;15:91.

Waldron L, Riester M. Meta-Analysis in Gene Expression Studies. *Methods Mol Biol*. 2016;1418:161-176.

Toro-Domínguez D, Villatoro-García JA, Martorell-Marugán J, Román-Montoya Y, Alarcón-Riquelme ME, Carmona-Sáez P. A survey of gene expression meta-analysis: methods and applications. *Brief Bioinform*. 2021;22(2):1694-1705.

Argelaguet R, Cuomo ASE, Stegle O, Marioni JC. Computational principles and challenges in single-cell data integration. *Nat Biotechnol*. 2021;10.1038/s41587-021-00895-7.

Pierre-Jean M, Deleuze JF, Le Floch E, Mauger F. Clustering and variable selection evaluation of 13 unsupervised methods for multi-omics data integration. *Brief Bioinform*. 2020;21(6):2011-2030.

- The structure of Figure 4 is not consistent with other figures.

Unlike the previous figure on single-target prediction, Figure 4 (now Figure 5 in the updated manuscript) contains results on the prediction of metabolic antiviral targets in combination with remdesivir and thus has a different format.

Thank you for sending us your revised manuscript. We have now heard back from the three reviewers who were asked to evaluate your study. As you will see the reviewers are satisfied with the modifications made and think that the study is now suitable for publication.

Before we can formally accept your manuscript, we would ask you to address the following editorial-level issues:

REFEREE REPORTS

Reviewer #1:

The reviewer really appreciated the extensive work from the authors to address my previous concern. Now, the authors have address my previous concern.

Reviewer #2:

I suggest the publication of the paper in its current form. The authors did an excellent work during the revision.

Reviewer #3:

The authors did a nice job with this revision. Analysis is comprehensive, figures are clear and illustrative. Well done.

2nd Authors' Response to Reviewers**29th Sep 2021**

The authors have made all requested editorial changes.

Thank you again for sending us your revised manuscript. We are now satisfied with the modifications made and I am pleased to inform you that your paper has been accepted for publication.

Corresponding Author Name: Sumit K. Chanda, Eytan Ruppin

Manuscript Number: MSB-2021-10260